# Discrete-Smoothness in Online Algorithms with Predictions

**Yossi Azar**
Tel Aviv University
azar@tau.ac.il

**Debmalya Panigrahi**
Duke University
debmalya@cs.duke.edu

**Noam Touitou**[*]
Amazon
noamtwx@gmail.com

## Abstract

In recent years, there has been an increasing focus on designing online algorithms with (machine-learned) predictions. The ideal learning-augmented algorithm is comparable to the optimum when given perfect predictions (*consistency*), to the best online approximation for arbitrary predictions (*robustness*), and should interpolate between these extremes as a smooth function of the prediction error. In this paper, we quantify these guarantees in terms of a general property that we call *discrete-smoothness* and achieve discrete-smooth algorithms for online covering, specifically the facility location and set cover problems. For set cover, our work improves the results of Bamas, Maggiori, and Svensson (2020) by augmenting consistency and robustness with smoothness guarantees. For facility location, our work improves on prior work by Almanza et al. (2021) by generalizing to nonuniform costs and also providing smoothness guarantees by augmenting consistency and robustness.

## 1 Introduction

The field of *learning-augmented online algorithms* has gained rapid prominence in recent years. The basic premise is to provide an online algorithm with additional (machine-learned) predictions about the future to help bypass worst-case lower bounds. Since machine-learned predictions can be noisy in general, a key desideratum of the model is that the *competitive ratio* of the online algorithm should degrade gracefully with prediction error. In particular, the cost of the algorithm should be bounded against that of the predicted solution (called *consistency*) or that of an online algorithm without predictions (called *robustness*) and should smoothly interpolate between the two with increase in prediction error (called *smoothness*). (The terms consistency and robustness were originally coined by Purohit, Svitkina, and Kumar [38].) While robustness and consistency are problem-independent notions, smoothness depends on prediction error which has been defined in a problem-specific manner. In this paper, we introduce a novel, problem-independent notion of smoothness called *discrete-smoothness* that applies to any combinatorial problem. As illustrative applications of this new framework, we design discrete-smooth (learning-augmented) algorithms for two classic problems, *facility location* and *set cover*, which improve and generalize previous results for these problems due to Almanza et al. (NeurIPS '21 [1]) and Bamas et al. (NeurIPS '20 [11]).

First, we introduce discrete-smoothness. Suppose we are given a problem instance of size $n$. Let OPT be a solution for this instance. (The reader may think of OPT as an optimal solution, although our guarantees will hold for any feasible solution.) Let the predicted solution be $S$. Ideally, $S = $ OPT; therefore, in general, OPT comprises two parts: the predicted part $\text{OPT}|_S := \text{OPT} \cap S$ and the unpredicted part $\text{OPT}|_{\overline{S}} := \text{OPT} \setminus S$. On the predicted part $\text{OPT}|_S$, the algorithm has a meaningful signal from the prediction but the noise in the signal is given by the overprediction $s_\Delta := |S \setminus \text{OPT}|$. Naturally, the competitive ratio of the algorithm on this part will degrade with increase in this noise. On the unpredicted part $\text{OPT}|_{\overline{S}}$, the algorithm does not have any signal from the prediction and

---

[*]This paper does not relate to the author's work at Amazon.

37th Conference on Neural Information Processing Systems (NeurIPS 2023).

cannot hope for a better competitive ratio than that of an online algorithm without prediction. Slightly abusing notation, we use $\text{OPT}|_S, \text{OPT}|_{\overline{S}}$ to denote both the aforementioned sets of items and their total cost; putting the two together, a learning-augmented algorithm ALG should satisfy

$$\text{ALG} \leq O(f(s_\Delta)) \cdot \text{OPT}|_S + O(f(n)) \cdot \text{OPT}|_{\overline{S}}, \tag{1}$$

where $O(f(\cdot))$ is the competitive ratio without prediction. We call the property of Equation (1) *discrete-smoothness*.

Let us first argue that Equation (1) recovers consistency and robustness. Consistency follows from setting $S = \text{OPT}$; then, Equation (1) demands a constant approximation to OPT. Similarly, robustness follows from the fact that for any $S$, the right hand side of Equation (1) is at most $O(f(n)) \cdot \text{OPT}$.

Next, we show that the two terms $f(s_\Delta)$ and $f(n)$ in Equation (1) are the best possible. For the first term, consider a prediction $S$ comprising the entire instance (of size $n$); in this case, we cannot hope for the better than $f(n)$-competitive algorithm; thus, $f(s_\Delta)$ is necessary in the first term. And, for the second term, consider an empty prediction $S = \emptyset$, in which case we again cannot hope for a better than $f(n)$-competitive algorithm; thus, $f(n)$ is necessary in the second term. Note that the asymmetry between these two terms is necessary: specifically, $f(n)$ cannot be replaced by $f(|\text{OPT} \setminus S|)$ since that would imply an $f(\text{OPT})$-competitive online algorithm when $S = \emptyset$. This is impossible, e.g., for the set cover problem.

A technical subtlety of the definition of discrete-smoothness (Equation (1)) is that given a fixed prediction $S$, the minimum value of the right hand side might actually be a solution OPT that is different from an optimal solution to the problem instance. So, although the solution OPT is intuitively an optimal solution, we require that a discrete-smooth algorithm satisfy Equation (1) for *all* feasible solutions OPT, and not just optimal solutions.

## 1.1 Our Results

We apply discrete-smoothness to the classic problems of online facility location and set cover. For these problems, we obtain results that improve on prior work. We describe these next.

**Online Facility Location with Predictions.** In the online facility location problem, we are given offline a metric space $(X, \delta)$ of $m := |X|$ points, where each point $v \in X$ has an associated facility opening cost $o_v \geq 0$. On receiving an online request for a client at some location $x \in X$, the online algorithm must connect the client to an open facility at some location $v \in X$ incurring connection cost $\delta(x, v)$. At any time, the algorithm is also allowed to open a facility at any location $v \in X$ by incurring the opening cost $o_v$. (Note that a client cannot update her connection even if a closer facility is opened later.) The total cost of the algorithm is the sum of opening costs of opened facilities and connection costs of clients.

The first result for the online facility location problem is due to Meyerson [33] who obtained a randomized algorithm with a competitive ratio of $O(\log n)$ for $n$ requests. This result was first derandomized [18], and later the competitive ratio slightly improved to $O\left(\frac{\log n}{\log \log n}\right)$ [19], by Fotakis. This latter bound is asymptotically tight. More recently, the online facility location problem has been considered in the context of machine-learned *predictions* (OFLP) by several papers [20, 1, 22]. Of these, the work of Almanza et al. [1] is the closest to our work (the other papers use metric error measures that are incomparable to our results). In [1], the offline input additionally contains a predicted solution of facilities $S \subseteq X$, where we denote $|S| = s$. By restricting the available facilities to the predicted set, they obtained an $O(\log s)$-competitive algorithm for uniform facility opening costs, under the condition that $\text{OPT} \subseteq S$.

We improve and generalize the Almanza et al. work by giving a discrete-smooth algorithm for the OFLP problem, i.e., an algorithm ALG that satisfies Equation (1):

**Theorem 1.1.** *There is an algorithm* ALG *for online (nonuniform) facility location with a predicted solution S that satisfies for every solution* OPT

$$\text{ALG} \leq O(\log s_\Delta) \cdot \text{OPT}|_S + O(\log n) \cdot \text{OPT}|_{\overline{S}}, \tag{2}$$

*where $s_\Delta$ is the number of facilities in $S \setminus \text{OPT}$ and $n$ is the number of online requests. Here, $\text{OPT}|_S$ (resp., $\text{OPT}|_{\overline{S}}$) represents the sum of opening costs of facilities in $\text{OPT} \cap S$ (resp., $\text{OPT} \setminus S$) and connection costs of all clients connecting to facilities in $\text{OPT} \cap S$ (resp., $\text{OPT} \setminus S$).*

This generalizes and improves the Almanza et al. result in three ways:

- The result is generalized from uniform facility opening costs to arbitrary (nonuniform) costs. In fact, even for the online facility location problem (without prediction), we get an $O(\log m)$-competitive algorithm for arbitrary (nonuniform) facility opening costs — previously, Almanza et al. only established this for uniform costs.

- The assumption that $\text{OPT} \subseteq S$, i.e., the prediction contains the entire solution, is no longer required.

- If $\text{OPT} \subseteq S$ (i.e., under the assumption of the Almanza et al. result), the competitive ratio improves from $O(\log s)$ to $O(\log s_\Delta)$. That is, the dependence is only on the prediction error and not the entire prediction.

In some situations, the length of the request sequence $n$ can exceed the size of the metric space $m$. To address this situation, we show that $n$ can be replaced by $m$ in the above result:

**Theorem 1.2.** *There is an algorithm* ALG *for online (nonuniform) facility location with a predicted solution S that satisfies for every solution* OPT

$$\text{ALG} \leq O(\log s_\Delta) \cdot \text{OPT}|_S + O(\log m) \cdot \text{OPT}|_{\overline{S}}, \tag{3}$$

*where m is the number of facilities in the metric space overall.*

**Online Set Cover with Predictions.** In the online set cover problem, we are given offline a universe of elements $E$ and $m$ sets defined on them $U \subseteq 2^E$ with nonnegative costs. In each online step, we get a new element $e \in E$. If $e$ is not already covered by the current solution, then the algorithm must add a new set from $U$ that contains $e$ to its solution. The total cost of the algorithm is the sum of costs of all sets in its solution.

Alon et al. [3] gave the first algorithm for the online set cover problem by introducing the online primal dual method, and obtained a competitive ratio of $O(\log m \log n)$ where $n$ denotes the number of requests. They also proved an almost matching lower bound of $\Omega\left(\frac{\log m \log n}{\log \log m + \log \log n}\right)$. Bamas, Maggiori, and Svensson [11] extended their work to online set cover *with predictions* (OSCP), where the offline input additionally contains a predicted solution of sets $S \subseteq U$. They established consistency and robustness bounds for this problem by adapting the online primal dual method to use the predicted solution. The cost of their algorithm is bounded by the minimum of $O(\log n)$ times the cost of the prediction and $O(\log m \log n)$ times the optimal cost. However, one cannot achieve smoothness through their work without choosing a trust parameter correctly in advance of the input.

We obtain a discrete-smooth algorithm for the OSCP problem, thereby giving the first algorithm for OSCP that goes beyond only consistency and robustness and achieves a smoothness guarantee:

**Theorem 1.3.** *There is an algorithm* ALG *for online set cover with a predicted solution S that satisfies for every solution* OPT

$$\text{ALG} \leq O(\log s_\Delta \log n) \cdot \text{OPT}|_S + O(\log m \log n) \cdot \text{OPT}|_{\overline{S}}, \tag{4}$$

*where $s_\Delta$ is the number of sets in $S \backslash \text{OPT}$. Here, $\text{OPT}|_S$ (resp., $\text{OPT}|_{\overline{S}}$) represents the sum of costs of sets in $\text{OPT} \cap S$ (resp., $\text{OPT} \backslash \overline{S}$).*

### 1.2 Our Techniques: A Framework for Discrete-Smooth Algorithms

At a high level, our framework merges two online algorithms to obtain a discrete-smooth algorithm. The algorithms differ in the guarantees they provide. The first algorithm $\text{ALG}_1$ gets a sharper competitive ratio of $O(f(s))$ but against the optimal solution restricted to the prediction $S$. The second algorithm $\text{ALG}_2$ has the standard competitive ratio of $O(f(n))$ but against the unconstrained optimum OPT. The main challenge in the combiner algorithm (call it ALG) is to decide how to route online requests to the two algorithms. The natural choice would be to decide this based on whether $\text{OPT}|_S$ or $\text{OPT}|_{\overline{S}}$ serves the request in OPT: in the first case, the request should be routed to $\text{ALG}_1$ and in the second case, it should be routed to $\text{ALG}_2$. But, of course, we do not know OPT and therefore don't know $\text{OPT}|_S$ and $\text{OPT}|_{\overline{S}}$.

Before we describe the combiner strategy, consider the properties that these algorithms need to satisfy.

- First, consider the *subset* of requests served by $\text{OPT}|_S$. Intuitively, $\text{ALG}_1$ should be competitive on these requests, which means that we need a stronger property from $\text{ALG}_1$ that its cost on any subset of requests is competitive against the optimal solution for this subset. We call this property *subset competitiveness*.[2] Symmetrically, subset competitiveness of $\text{ALG}_2$ ensures that it is competitive on the requests in $\text{OPT}|_{\overline{S}}$.

- Next, we need a guarantee on the cost of $\text{ALG}_1$ on $\text{OPT}|_{\overline{S}}$, and symmetrically, of $\text{ALG}_2$ on $\text{OPT}|_S$. For this, we first augment $\text{ALG}_1, \text{ALG}_2$ to address the *prize-collecting* version of the original problem, where each online request can be ignored at a *penalty cost*. (Note that this is more general than the original problem where every online request must be served, since the latter can be recovered by setting the penalties to be infinitely large.) Setting the penalties appropriately, we ensure that the total penalty of the requests in $\text{OPT}|_S$ is bounded against the cost of $\text{ALG}_1$ on those requests (similarly for $\text{OPT}|_{\overline{S}}$).

- Finally, we require that the cost of $\text{ALG}_1, \text{ALG}_2$ on any set of requests is bounded against the total penalty of the requests. We call this strengthened competitiveness w.r.t. penalties the *Lagrangian property*[3]. Note that this ensures that the cost of $\text{ALG}_1, \text{ALG}_2$ on $\text{OPT}|_{\overline{S}}, \text{OPT}|_S$ are respectively bounded.

Now, we give the formal definition of Lagrangian subset-competitiveness that we motivated above. We use $\text{ALG}(Q'|Q)$ to refer to the total cost of ALG incurred when addressing a subset $Q' \subseteq Q$ as part of running on an input $Q$. For any prize collecting solution SOL for input $Q$, we separate its total cost into $\text{SOL}^b(Q)$ (buying cost) and $\text{SOL}^p(Q)$ (penalty cost). We formalize the Lagrangian subset-competitiveness property below:

**Definition 1.4 (Lagrangian subset-competitive algorithm).** Let ALG be a randomized prize-collecting algorithm running on an input $Q$. For any competitive ratio $\beta$, we say that ALG is Lagrangian $\beta$-subset-competitive if for every subset $Q' \subseteq Q$ we have

$$\mathbb{E}[\text{ALG}(Q'|Q)] \le \beta \cdot \text{OPT}^b(Q') + O(1) \cdot \text{OPT}^p(Q') \tag{5}$$

If in the equation above we replace the unconstrained optimum (OPT) by the optimal solution that can only use the prediction $S$, we say that ALG is Lagrangian $\beta$-subset-competitive w.r.t. $S$.

We now give the combiner algorithm:

---

**Algorithm 1:** Smooth merging framework (The combiner algorithm)

---

1 Let $\text{ALG}_1, \text{ALG}_2$ be two prize-collecting Lagrangian subset-competitive algorithms.
2 **Event Function** UPONREQUEST($q$)
3    Let $\alpha$ be the minimum penalty such that releasing $(q, \alpha)$ to $\text{ALG}_1, \text{ALG}_2$ would result in the request being served in either $\text{ALG}_1$ or $\text{ALG}_2$. (The value of $\alpha$ can be determined by a standard "guess-and-double".)
4    Release $(q, \alpha)$ to both $\text{ALG}_1$ and $\text{ALG}_2$. Buy the items bought by $\text{ALG}_1, \text{ALG}_2$ as a result of this step.

---

The algorithm is simple: for a new online request $q$, the framework chooses the minimum penalty $\alpha$ which ensures that at least one of the two constituent algorithms $\text{ALG}_1, \text{ALG}_2$ would actually serve $q$ (instead of paying the penalty). $(q, \alpha)$ is then presented as a (prize-collecting) request to both algorithms. (Recall that the combined algorithm is for the non-prize-collecting problem, but the individual algorithms $\text{ALG}_1, \text{ALG}_2$ are for the prize-collecting problem.) At this stage, one of the algorithms serves the request (due to the choice of $\alpha$) while the other may choose to pay the penalty. The combiner algorithm now simply buys all items bought by either algorithm.

Finally, we state the guarantees of the combiner algorithm informally. (For a formal description, see Appendix C.)

**Theorem 1.5.** *(Informal) If* $\text{ALG}_1, \text{ALG}_2$ *are Lagrangian $\beta$-subset-competitive algorithms for $\beta = f(s), f(n)$ respectively, then Algorithm 1 satisfies the discrete-smoothness property (Equation* (1)*.*

---

[2]Our subset-competitiveness property is similar to [9].

[3]Our Lagrangian competitiveness is similar to the Lagrangian multiplier preserving property in approximation algorithms for prize-collecting problems, e.g., [37, 26].

**Applications of Theorem 1.5:** Section 3 and Appendix B give Lagrangian subset-competitive algorithms for facility location, and Section 4 gives a Lagrangian subset-competitive algorithm for set cover. Given these constituent algorithms, we use Theorem 1.5 to prove Theorem 1.1 and Theorem 1.2 for facility location and Theorem 1.3 for set cover. These proofs are given in Appendix D.

**Related Work.** There is a growing body of work in online algorithms with predictions in the last few years (see, e.g., the surveys [35, 36]). This model was introduced by Lykouris and Vassilvitskii for the caching problem [32] and has since been studied for a variety of problem classes: rent or buy [27, 25, 21, 41, 5, 39], covering [11], scheduling [27, 41, 10, 28, 34, 30, 8], caching [31, 40, 24, 13], matching [29, 16, 7, 23], graph problems [6, 22, 1, 14, 4, 20, 9], and so on. Prior works on online facility location with predictions either do not consider prediction error [1] or use continuous notions of error [22, 20], such as functions of the distances between predicted and optimal facilities. Our discrete notion of error refers only to whether an optimal item is predicted. Similarly, prior work on online set cover with predictions [11, 4] also does not consider prediction error. Finally, we note that discrete prediction error (similar to this paper) as well as hybrids between discrete and continuous error have also been considered [42, 9, 14] but the prediction here is on the input rather than the solution.

## 2 The Framework

We now describe some of the concepts of the framework in more detail.

**Reduction from $s_\delta$ to $s$.** Recall that we seek discrete-smooth algorithms, i.e., satisfying Equation (1). Our first step is to give a generic reduction that allows us to slightly weaken the guarantee to the following:

$$\text{ALG} \leq O(f(s)) \cdot \text{OPT}|_S + O(f(n)) \cdot \text{OPT}|_{\overline{S}}, \tag{6}$$

where $O(f(\cdot))$ is the competitive ratio without predictions. We give a reduction from an algorithm that satisfies Equation (6) to one that satisfies Equation (1):

**Theorem 2.1.** *Given an algorithm* $\text{ALG}'$ *such that* $\text{ALG}' \leq O(f(s)) \cdot \text{OPT}|_S + O(g) \cdot \text{OPT}|_{\overline{S}}$, *there exists an algorithm* ALG *such that* $\text{ALG} \leq O(f(s_\Delta)) \cdot \text{OPT}|_S + O(g) \cdot \text{OPT}|_{\overline{S}}$.

The proof of this theorem, in Appendix E, is roughly the following: for every integer $i$, once the cost of the algorithm exceeds $2^i$, we buy the cheapest predicted items of total cost at most $2^i$, and then remove them from the prediction. While $2^i < \text{OPT}$, the total cost is $O(1) \cdot \text{OPT}$; once $2^i$ exceeds OPT, the size of the prediction is at most $s_\Delta$, and Equation (6) implies Equation (1).

**Monotonicity.** An additional, natural property that we demand from a constituent algorithm in our smooth combination framework is that increasing the penalty of input requests does not decrease the cost incurred by the algorithm. This is stated formally in the following definition.

**Definition 2.2.** We say that a prize-collecting algorithm ALG is *monotone* if, fixing the input request prefix $((q_i, \pi_i))_{i=1}^{k-1}$ and current request $(q_k, \pi_k)$, then increasing $\pi_k$ does not decrease $\text{ALG}(q_k, \pi_k)$.

**Online amortization.** Our framework extends to the case where Lagrangian subset-competitiveness and monotonicity are satisfied by *amortized* costs instead of actual costs. This is important because for some problems, the actual cost expressly prohibits subset competitiveness. For example, consider facility location: given an input of multiple, identical requests with very small penalty, the algorithm should eventually stop paying penalties and open a facility. However, for the specific request upon which the facility is opened, the cost of the algorithm is much larger than the penalty for that request, the latter being optimal for just that request. To overcome this complication, we allow the cost for a request to be amortized over previous requests, and call this *online amortization*.

First, we define online amortization of costs, and define a "monotone" online amortization which can be used in our framework.

**Definition 2.3** (online amortization). Let $Q = ((q_1, \pi_1), \cdots, (q_n, \pi_n))$ be an online input given to ALG. An *online amortization* or OA is a number sequence $(\text{OA}(q, \pi))_{(q, \pi) \in Q}$ such that:

1. $\text{ALG}(Q) \leq \sum_{(q,\pi) \in Q} \text{OA}(q, \pi)$.

2. $\text{OA}(q_i, \pi_i)$ is only a function of $(q_1, \pi_1), \cdots, (q_i, \pi_i)$, and can thus be calculated online.

When considering the amortized cost of an algorithm, we use similar notation to the actual cost: on an input $Q$, we use $\text{OA}(Q)$ to denote the total amortized cost. We also use $\text{OA}(Q'|Q)$ to denote the total amortized cost incurred on a request subset $Q' \subseteq Q$. In addition, for a request $(q, \pi)$ in the input $Q$, we use $\text{OA}(q, \pi)$ to refer to the amortized cost of $(q, \pi)$; here the input $Q$ is clear from context.

**Definition 2.4** (monotone online amortization)**.** We call an online amortization $\text{OA}$ *monotone* if **(a)** fixing previous requests, increasing the penalty of request $(q, \pi)$ never decreases $\text{OA}(q, \pi)$, and **(b)** when the algorithm pays penalty for $(q, \pi)$ then $\text{OA}(q, \pi) \geq \pi$.

**The Main Theorem**   We are now ready to state the main theorem of our algorithmic framework. We use $\beta_1$ and $\beta_2$ to denote the competitive ratios of $\text{ALG}_1$ and $\text{ALG}_2$; the reader should think of $\beta_1$ as $O(f(s))$ and $\beta_2$ as $O(f(n))$, i.e., $\beta_2 \gg \beta_1$.

**Theorem 2.5.** *Consider any online covering problem with predictions $\mathcal{P}$. Let $\text{ALG}_1, \text{ALG}_2$ be two algorithms for the prize-collecting version of $\mathcal{P}$ with monotone (online amortized) costs $\text{OA}_1, \text{OA}_2$ respectively such that **(a)** $\text{ALG}_1$ is Lagrangian $\beta_1$-subset-competitive using $\text{OA}_1$ w.r.t. the prediction $S$, and **(b)** $\text{ALG}_2$ is Lagrangian $\beta_2$-subset-competitive using $\text{OA}_2$ (against general $\text{OPT}$).*

*Then there exists $\text{ALG}$ for $\mathcal{P}$ such that for every partition of the input $Q$ into $Q_1, Q_2$ we have*

$$\text{ALG}(Q) \leq O(\beta_1) \cdot \text{OPT}_S(Q_1) + O(\beta_2) \cdot \text{OPT}(Q_2)$$

We later show that Theorem 2.5 implies Equation (6) for facility location and set cover.

## 3   Online Facility Location

In this section, we consider metric, nonuniform facility location with predictions and present a novel prize-collecting algorithm TREEPROXY. This algorithm is Lagrangian $O(\log|S|)$-subset-competitive w.r.t. the prediction $S$ of possible facilities; thus, it is used in our framework to prove Theorems D.2 and D.3, which in turn imply Theorems 1.1 and 1.2, respectively. In addition, TREEPROXY is a result independent of our framework/predictions: the competitiveness guarantee shown for TREEPROXY also achieves $O(\log m)$ competitiveness where $m = |X|$ is the size of the metric space. We prove the following theorem:

**Theorem 3.1.** *For facility location with predictions, there exists a randomized prize-collecting algorithm $\text{ALG}$ with a monotone online amortization $\text{OA}$ which is Lagrangian $O(\log|S|)$-subset competitive using $\text{OA}$ w.r.t. $S$.*

### 3.1   The Algorithm

**Weighted hierarchically-separated trees (HSTs).** The algorithm starts by embedding the metric space into the leaves of a weighted 3-HST, a metric space in which edge weights decrease at least exponentially as one descends from the root.

**Definition 3.2.** For $\gamma > 1$, a rooted tree with weights $c$ to the edges is a *weighted $\gamma$-HST* if for every two edges $e_1, e_2$ such that $e_2$ is a parent edge of $e_1$, it holds that $c(e_2) \geq \gamma c(e_1)$.

The following result is often used for embedding general metric spaces into weighted HSTs; it involves composing the embeddings of Fakcharoenphol et al. [17] and Bansal et al. [12].

**Theorem 3.3** (Due to [17] and [12])**.** *For every metric space $(X, \delta)$ and constant $\gamma$, there exists a distribution $\mathcal{D}$ over weighted $\gamma$-HSTs of depth $O(\log|X|)$ in which the points in $X$ are the leaves of the HST, such that for every two points $x_1, x_2 \in X$ we have:*

1. *$\delta(x_1, x_2) \leq \delta_T(x_1, x_2)$ for every $T$ in the support of $\mathcal{D}$.*

2. *$\mathbb{E}_{T \sim \mathcal{D}}[\delta_T(x_1, x_2)] \leq O(\log|X|) \cdot \delta(x_1, x_2)$.*

The algorithm starts by embedding the induced metric space of $S$ into a weighted HST using Theorem 3.3; $T$ denotes the resulting tree, and $r$ denotes its root. For each edge $e \in T$, we denote by $c(e)$ the cost of the edge $e$. Denote the set of leaves in the subtree rooted at $v$ by $L(v)$; note that $L(r) = S$. Denote the distance between two nodes $u, v$ in the tree by $\delta_T(u, v)$. For every point $u \in X$, define $p(u) := \arg\min_{u' \in S} \delta(u, u')$; that is, $p(u)$ is the closest predicted point to $u$ (abusing notation, we similarly define $p(q)$ for request $q$).

**Proxy list.** After embedding $S$ into the leaves of a tree, the algorithm must open facilities on those leaves to serve requests. Intuitively, at any point the algorithm considers some (possibly internal) node $v \in T$, and considers connecting the current request through $v$ to a facility in $L(v)$. Choosing from $L(v)$ introduces a tradeoff between the cost of opening the facility and its distance from $v$. For every $v$, we identify the leaves in $L(v)$ which offer the best points in this tradeoff (i.e., a Pareto frontier), and only allow the algorithm to choose from these leaves. This subset is called the *proxy list* of $v$, and denoted $P(v) \subseteq L(v)$.

We now define the proxy list $P(v)$. For ease of notation, define the logarithmic class operator $\ell(x) := \lfloor \log x \rfloor$. For node $v \in T$, we construct the proxy list $P(v) \subseteq L(v)$ as follows:

1. Start with $V \leftarrow L(v)$.

2. While there exist distinct $v_1, v_2 \in V$ such that $\ell(o_{v_1}) \geq \ell(o_{v_2})$ and $\ell(\delta_T(v, v_1)) \geq \ell(\delta_T(v, v_2))$, remove $v_1$ from $V$.

3. Output $V$ as $P(v)$.

We denote by $k(v)$ the size of the proxy list $P(v)$. We order the proxy list of $v$ by increasing facility cost, thus writing $P(v) = (s_1^v, \cdots, s_{k(v)}^v)$. For every $v, i$, we use the shorthands $o_i^v := o_{s_i^v}$ and $\delta_i^v := \delta_T(v, s_i^v)$. Slightly abusing notation, for every node $v \in T$ we define $c(v) := c(e_v)$ where $e_v$ is the edge connecting $v$ to its parent node (for $r$, we define $c(r) = \infty$). For a more streamlined notation, for every node $v \in T$ we define $\delta_0^v := c(v)$ and $o_{k(v)+1}^v := \infty$.

**Observation 3.4.** *For every node $v \in T$, the proxy list $P(v)$ satisfies:*

1. *For every $u \in L(v)$, there exists index $i$ such that $\ell(o_i^v) \leq \ell(o_u)$ and $\ell(\delta_i^v) \leq \ell(\delta_T(v, u))$.*

2. *For every distinct $1 \leq i < j \leq k(v) + 1$, it holds that $\ell(o_i^v) < \ell(o_j^v)$.*

3. *For every distinct $0 \leq i < j \leq k(v)$, it holds that $\ell(\delta_i^v) > \ell(\delta_j^v)$.*

When $i = 0$, the third item in Observation 3.4 uses the fact that $T$ is a weighted 3-HST; thus, the cost of an edge is at least twice the distance from the child node of that edge to any descendant leaf.

**Counters.** For every node $v$ and every $i \in \{1, \cdots k(v) + 1\}$, we define a counter $\lambda(v, i)$ of size $o_i^v$.

**Algorithm description.** The algorithm for facility location with predictions is given in Algorithm 2. Initially, the algorithm embeds the metric space induced by $S$ into a weighted 3-HST $T$, using Theorem 3.3; upon each node in this $T$ the proxy lists are computed, and the corresponding counters are assigned. Upon the release of a request $(q, \pi)$, the function UPONREQUEST is triggered. Upon receiving $(q, \pi)$, it maps the request to the closest point $p(q)$ in $S$ (that is, a leaf of the HST). Then, the algorithm attempts to solve the request on the HST through a process of increasing counters, which we soon describe. (While the described algorithm raises these counters continuously, the process can easily be discretized, replacing the continuous growth with jumping discretely to the next event.) The algorithm keeps track of (some measure of) the cost involved; if during UPONREQUEST that amount exceeds the penalty $\pi$, the algorithm pays the penalty instead (see Line 9).

When solving the request on $u = p(q)$, the algorithm climbs up the branch of $u$, until a facility is found (or opened) to connect $u$. At each ancestor $v$ of $u$, the algorithm invests a growing amount $\tau_v$ in advancing the proxy list of $v$ (i.e., buying a facility in $P(v)$ closer to $v$). It raises the counter for the next item on the proxy list until full, at which point the relevant proxy facility is opened, and the next counter in the proxy list begins to increase. (Note that the same facility can be "opened" more than once due to being on multiple proxy lists.) Once $\tau_v$ reaches the cost of connecting $v$ to an open proxy, the algorithm stops increasing counters and makes the connection. When no proxy in $P(v)$ is open, it could be that $\tau_v$ exceeds the cost of moving from $v$ to its parent $p(v)$; in this case, we ascend the branch and explore proxies for $p(v)$. Note that the function UPONREQUEST of Algorithm 2 also returns a value; this return value is the online amortization cost of the request, to be used in the analysis of the algorithm. (See Figure 1 for an example.)

The analysis of Algorithm 2, and the proof of Theorem 3.1, appear in Appendix A.

**Algorithm 2:** TREEPROXY for Prize-Collecting Facility Location with Predictions

**1 Initialization**
**2** Embed the prediction $S$ into a weighted 3-HST $T$ using Theorem 3.3.
**3** For every $v \in T$, and every $i \in \{1, \cdots, k(v) + 1\}$, set $\lambda(v, i) \leftarrow 0$.
**4** For every $v \in T$, set $t(v) \leftarrow 0$.

**5 Event Function** UPONREQUEST$(q, \pi)$
  *// Upon the next request $q$ with penalty $\pi$ in the sequence*
**6** Define $u, v \leftarrow p(q)$.
**7** Define $\tau \leftarrow 0, \tau^v \leftarrow 0$.
**8** **continually increase** $\tau$, $\tau^v$ *and* $\lambda(v, t(v) + 1)$ *at the same rate* **until:**
**9** if $\tau + \delta(u, q) \geq \pi$ **then** *// cost for request exceeds penalty; pay penalty instead.*
**10** Pay the penalty $\pi$ for the request.
**11** **return** $\tau + \pi$. *// return amortized cost.*
**12** if $\lambda(v, t(v) + 1) = o^v_{t(v)+1}$ **then** *// counter for next proxy is full; open facility at proxy.*
**13** Open a facility at $s^v_{t(v)+1}$.
**14** Increment $t(v)$.
**15** **goto** Line 8.
**16** if $\tau^v \geq \delta^v_{t(v)}$ **then**
**17** if $t(v) = 0$ **then**
  *// escalate the request to parent node.*
**18** Set $v \leftarrow p(v)$.
**19** Define $\tau^v \leftarrow 0$.
**20** **goto** Line 8.
**21** Connect $q$ to $s^v_{t(v)}$. *// connect request to closest proxy.*
**22** **return** $\tau + (\tau + \delta(u, q))$. *// return amortized cost.*

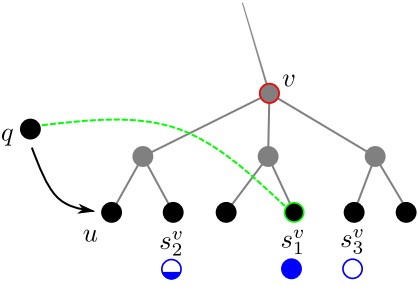

Figure 1: A possible state of Algorithm 2, immediately before connecting a request $q$. Here, $q$ has been mapped to $u$, which is the closest point in $S$. The variable $v$, an ancestor of $u$, is shown, as is its proxy list $s^v_1, s^v_2, s^v_3$. The counters of the proxy list are also shown: $\lambda(v, 1)$ is full (and a facility thus exists in $s^v_1$), and $\lambda(v, 2)$ is partial (the last counter to be raised handling $q$). At some point, the growth in the counters of $v$ exceeded the distance from $v$ to $s^v_1$, and thus the connection of $q$ to $s^v_1$ is made.

**Algorithm 3:** Online Prize-Collecting Fractional Set Cover

**1 Initialization**
**2** Set $x_s \leftarrow 0$ for every set $s$.

**3 Event Function** UPONREQUEST $(q, \pi)$
**4** Set $y_q \leftarrow 0$.
**5** **while** $\sum_{s \in U(q)} x_s \leq 1$ **do**
**6** Set $y_q \leftarrow y_q + 1$
**7** if $\pi \leq y_q$ **then**
**8** Pay penalty $\pi$ for $q$.
**9** **return** OA$(q, \pi) = 3\pi$.
**10** **foreach** $s \in U(q)$ **do**
**11** $x_s \leftarrow x_s \cdot (1 + \frac{1}{c_s}) + \frac{1}{|U(q)|c_s}$
**12** **return** OA$(q, \pi) = 2y_q$.

## 4 Online Set Cover

In this section, we present and analyze an algorithm for prize-collecting fractional set cover which uses the well-known multiplicative updates method, and show that it is Lagrangian subset-competitive. Using this algorithm together with Algorithm 1 yields Theorem 1.3 (the proof appears in Appendix C).

*Preliminaries.* In prize-collecting fractional set cover, we are given a universe with elements $E$ and sets $U$; we define $m := |U|$. A solution may fractionally buy sets, according to a cost function $c$. Requests then arrive online, where each request is for covering some element $e \in E$, which is contained in some subfamily of sets from $U$. To cover an element, an algorithm must hold fractions of sets containing $e$ which sum to at least 1. Observe that fractional set cover with predictions conforms to the definition of an online covering problem with predictions; in this problem, the items are the sets. For prize-collecting fractional set cover, we prove the following theorem.

**Theorem 4.1.** *There exists a deterministic algorithm* ALG *for prize-collecting fractional set cover that* ALG *is Lagrangian $O(\log m)$-subset-competitive*

Theorem 4.1 implies that, in the framework of Algorithm 1, our algorithm can be used as the general component, independent of the prediction. But, given a prediction $S \subseteq U$, we can simply restrict the

family of sets used by the algorithm to the given prediction, yields an algorithm competitive against $\text{OPT}_S$. Thus, Theorem 4.1 immediately yields the following corollary.

**Corollary 4.2.** *There exists a deterministic algorithm* ALG *for prize-collecting fractional set cover such that* ALG *is Lagrangian $O(\log m')$-subset-competitive w.r.t. prediction $S \subseteq U$, where $|S| = m'$.*

**The Algorithm.** The algorithm for prize-collecting set cover is given in Algorithm 3. The algorithm follows the standard multiplicative updates method: while the pending request is uncovered, sets containing that request are bought at an exponential rate (see [2, 15]). However, in this prize-collecting version, the algorithm never lets its cost for a specific request exceed its penalty. For ease of notation, define $U(q)$ to be the collection of sets containing $q$; that is, $U(q) := \{s \in U | q \in s\}$.

**Analysis.** Where the input $Q$ is fixed, and for $(q, \pi) \in Q$, we use $\text{ALG}(q, \pi)$ as a shorthand for $\text{ALG}(\{(q, \pi)\} | Q)$; i.e., the cost of ALG when handling the request $(q, \pi)$ as part of $Q$. We prove the two following lemmas:

**Lemma 4.3.** *For every $(q, \pi) \in Q$, it holds that $\text{ALG}(q, \pi) \le 3\pi$.*

**Lemma 4.4.** *For every subset $Q' \subseteq Q$, we have $\text{ALG}(Q' | Q) \le O(\log m) \cdot \text{OPT}(\overline{Q'})$, where $\overline{Q'}$ is the non-prize-collecting input formed from $Q'$.*

These two lemmas imply *penalty-robust subset competitiveness*, a property shown in Proposition C.2 to be equivalent to Lagrangian subset-competitiveness. Thus, we focus on proving these lemmas; note that the proof of Lemma 4.4 appears in Appendix F.

**Proposition 4.5.** *In every iteration of* UPONREQUEST$(q, \pi)$*, it holds that the total buying cost is at most $2y_q$, where $y_q$ is the final value of the variable of the same name.*

*Proof.* Consider each time $y_q$ is incremented. The total cost of buying sets is the following.

$$\sum_{s \in U(q)} c_s \cdot \left( x_s \cdot \frac{1}{c_s} + \frac{1}{|U(q)| c_s} \right) = 1 + \sum_{s \in U(q)} x_s \le 2$$

where the inequality is due to the fact that $\sum_{s \in U(q)} x_s \le 1$. Thus, each time $y_q$ is incremented by 1, the cost of buying sets is at most 2, completing the proof. □

*Proof of Lemma 4.3.* Consider UPONREQUEST$(q, \pi)$. If it returned through Line 11, it holds that $y_q \le \pi$; Proposition 4.5 shows that the total buying cost was thus at most $2\pi$, and this cost is also $\text{ALG}(q, \pi)$. Otherwise, the function returned through Line 8; in this case, since $y_q$ was incremented immediately before comparing $y_q$ to $\pi$, the argument from the proof of Proposition 4.5 implies that the total buying cost is at most $2(y_q - 1)$ (using the final value of $y_q$). In turn, this is at most $2\pi$. In addition, the algorithm paid the penalty of $\pi$; overall, $\text{ALG}(q, \pi) \le 3\pi$. □

*Proof of Theorem 4.1.* Lemma 4.3 and Lemma 4.4 show that the algorithm is $O(\log m)$-PRSC; Proposition C.2 then yields that the algorithm is Lagrangian $O(\log m)$-subset-competitive. □

## 5 Experiments

*Input Generation.* Our set cover instances contain 100 elements. (The number of sets will vary in the experiments.) Every set contains every element with some constant probability $\alpha$ (we choose $\alpha = 0.02$); that is, the input is represented by a random bipartite graph in which each edge manifests independently. Since this may not cover every element, we also add singleton sets for all elements. We generate random costs for the sets, independently drawn from a log-normal distribution ($\mu = 0, \sigma = 1.6$). For a given input, we generate a prediction in the following way:

1. Using an LP solver, we obtain an optimal fractional solution to the problem instance.
2. We randomly round the solution, such that every set appears in the prediction with probability proportional to its value in the fractional solution.
3. We apply noise to the prediction, of two types: false-positive noise, in which every set is added to the prediction with some probability $p$; and false-negative noise, in which every set is removed from the prediction with some probability $q$. (The reader should think of $p$ and $q$ as the classification error where the predictions were generated using a classifier.)

| $p, q$ | ON comp. ratio | PREDON comp. ratio | BASEMERGE comp. ratio | SMOOTHMERGE comp. ratio |
|---|---|---|---|---|
| 0, 0 | 6.007 (0.244) | 1.689 (0.070) | 3.102 (0.565) | 2.779 (0.122) |
| 0, 0.15 | 6.007 (0.244) | 46.815 (54.436) | 6.246 (1.516) | 3.820 (0.555) |
| 0, 0.3 | 6.007 (0.244) | 96.156 (76.196) | 7.093 (1.358) | 4.824 (0.687) |
| 0.005, 0 | 6.007 (0.244) | 1.989 (0.106) | 3.648 (0.630) | 3.251 (0.184) |
| 0.005, 0.15 | 6.007 (0.244) | 25.983 (30.294) | 6.597 (1.642) | 4.200 (0.534) |
| 0.005, 0.3 | 6.007 (0.244) | 51.533 (43.375) | 7.543 (1.541) | 5.120 (0.642) |
| 0.02, 0 | 6.007 (0.244) | 2.631 (0.154) | 4.489 (0.660) | 4.240 (0.266) |
| 0.02, 0.15 | 6.007 (0.244) | 10.555 (7.549) | 7.007 (1.496) | 5.024 (0.498) |
| 0.02, 0.3 | 6.007 (0.244) | 17.588 (9.549) | 8.156 (1.433) | 5.760 (0.569) |

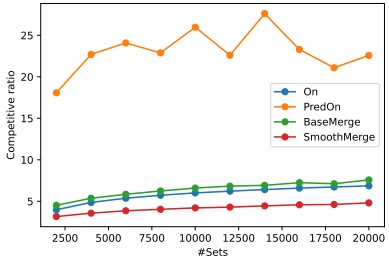

Table 1: Competitive ratios for varying $p, q$, in a "mean (standard deviation)" format. Best values in each row are underlined.

Figure 2: The competitive ratio for varying numbers of sets.

4. Finally, we add the singleton sets to the prediction, to ensure that the prediction covers all elements.

*Baselines and evaluation.* We evaluate our algorithm described in Section 4, denoted SMOOTHMERGE, against three baselines: the standard online algorithm without predictions, denoted ON; the online algorithm restricted to predicted sets, denoted PREDON; and the standard merging BASEMERGE of those two algorithms, which alternates between ON and PREDON whenever the overall cost doubles. For every choice of parameters, we measure the costs of the four algorithms; these costs are then averaged over 300 different random inputs. We then measure the expected competitive ratio of each algorithm. Our experiments were run on an AWS EC2 r5.16xlarge machine.

We ran the following experiments: (a) we vary the false-positive rate $p$ and the false-negative rate $q$ keeping the number of sets fixed at 10000 (Table 1), and (b) we vary the number of sets in the input, fixing $p = 0.005$, $q = 0.15$ (Figure 2).

*Experimental Results.* We ran two sets of experiments. In the first experiment, we varied the false-positive rate $p$ and the false-negative rate $q$ keeping the number of sets fixed at 10000. The results are reported in Table 1. We note that our algorithm SMOOTHMERGE outperforms the standard merging algorithm BASEMERGE and the online algorithm without predictions ON consistently across all values of $p, q$. SMOOTHMERGE also outperforms PREDON, the online algorithm restricted to the prediction, except when there are no false negatives, i.e., $q = 0$. This is to be expected because $q = 0$ implies that there is a good solution contained in the prediction. When $q > 0$, PREDON fails miserably and our algorithm SMOOTHMERGE obtains a competitive ratio that is an order of magnitude better than PREDON. This demonstrates the lack of robustness of PREDON because it is specifically tuned to correct predictions.

In the second set of experiments, we varied the number of sets in the input fixing the noise rates $p = 0.005$, $q = 0.15$. The results are reported in Figure 2. Our algorithm SMOOTHMERGE consistently outperforms all the baseline algorithms. In particular, it is able to utilize predictions to outperform ON, which the standard merging BASEMERGE is unable to achieve. Moreover, as the number of sets in the input grows, the gap between the two merging solutions increases.

# 6  Discussion

In this paper, we presented a novel framework for smooth interpolation between robustness and consistency guarantees in learning-augmented online algorithms, and applied it to set cover and facility location. More broadly, predictions for online algorithms are of two forms: prediction of the input and that of the solution. The notion of discrete-smoothness applies to any online combinatorial problem in the latter category, i.e., where a solution is provided in the form of a prediction to the algorithm. Many problems have been considered in this model including rent or buy problems, scheduling, matching, graph problems, etc. For all of these problems, the discrete-smoothness framework alleviates the need for problem-specific notions of prediction error and instead gives a common framework for arguing about the gradual degradation of solution quality with increase in prediction error. We hope that the current work will streamline the desiderata for learning-augmented online algorithms by adding this problem-independent notion of smoothness to the established (and also problem-independent) properties of consistency and robustness.

## Acknowledgments

YA was supported in part by the Israel Science Foundation (grant No. 2304/20). DP was supported in part by NSF awards CCF-1750140 (CAREER) and CCF-1955703.

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

# A Analysis of Algorithm 2

For this analysis section, we fix any input $Q = ((q_1, \pi_1), \cdots, (q_n, \pi_n))$. For both ALG and $\text{OPT}_S$ we use the superscript f to refer only to facility opening costs and c to refer only to connection costs. We denote by $\text{OA}(q, \pi)$ the value returned by UPONREQUEST in Algorithm 2 upon receiving the pair $(q, \pi)$; we choose $(\text{OA}(q, \pi))$ as the online amortization of Algorithm 2.

**Online Amortization.** First, we show that the cost of the algorithm is bounded by the online amortization:

**Lemma A.1.** *It holds that* $\text{ALG}(Q) \leq \text{OA}(Q)$.

*Proof.* We use the subscript $q$ to refer to the final value of a variable in UPONREQUEST$(q, \pi)$. The cost of the algorithm has the following three components:

1. Penalties paid.

2. Opening costs for facilities in $S$.

3. Connection costs for facilities in $S$.

Let $Q' \subseteq Q$ be the set of requests served by the algorithm (i.e., no penalty was paid).

**Penalties for requests in $Q \setminus Q'$.** Consider that whenever a penalty $\pi$ is paid for a request in Line 10, the additive term $\pi$ appears in the amortized cost of that request. We charge the penalty cost to that term.

**Opening cost.** Note that a facility $s_i^v$ is only opened (at cost $o_i^v$) when the counter $\lambda(v, i)$ reaches $o_i^v$, and that counter is never used again; thus, the total opening cost can be charged to the sum over request $q$ of the amount by which request $q$ raises counters, which is $\tau_q$. We charge this to the term $\tau_q$ in $\text{OA}(q, \pi)$.

**Connection cost for requests in $Q'$.** Suppose a request $(q, \pi) \in Q'$ is connected to some point $w \in S$. There exists an index $i$ such that $w = s_i^{v_q}$. It holds that

$$\delta(q, w) \leq \delta(q, u_q) + \delta(u_q, w) \leq \delta(q, S) + \delta_T(u_q, w) \leq \delta(q, S) + \delta_T(u_q, v_q) + \delta_T(v_q, w). \quad (7)$$

where the first and third inequalities are due to the triangle inequality, and the second inequality is due to the definition of $u_q$ and Theorem 3.3. Now, note that $\delta_T(v_q, w) = \delta_i^{v_q} \leq \tau_q^{v_q}$ from the condition of Line 16.

Enumerate the path in the tree between $u_q$ and $v_q$ as $u_q = w_0, w_1, \cdots, w_k = v_q$, and note that $\delta_T(u_q, v_q) = \sum_{j=0}^{k-1} c(w_j)$. Now, note that the variable $v$ advanced from $w_j$ to $w_{j+1}$ due to $\tau_q^{w_j} \geq c(w_j)$; thus, $\delta_T(u_q, v_q) = \sum_{j=0}^{k-1} \tau_q^{w_j}$. Finally, note that $\sum_{j=0}^{k} \tau_q^{w_j} = \tau_q$; combining, we get

$$\delta_T(u_q, v_q) + \delta_T(v_q, w) \leq \sum_{j=0}^{k} \tau^{w_j} = \tau_q$$

Plugging the above into Equation (7), we get $\delta(q, w) \leq \tau_q + \delta(q, S)$. We thus charge the connection cost of requests from $Q'$ to the $(\tau_q + \delta(q, S))$ term in $\text{OA}(q, \pi)$.

This completes the proof of the lemma. $\qquad \square$

**Observation A.2.** *The online amortization* OA *of Algorithm 2 is monotone.*

**Bounding Amortized Costs.** Having shown that the online amortization is valid and monotone, it remains to bound the amortized cost of the algorithm. To show that the algorithm is Lagrangian subset-competitive, it is enough to show that it is PRSC; see Proposition C.2. We thus focus on showing that the algorithm is PRSC using OA w.r.t. $S$.

From this point on, for every node $v \in T$ and index $i \in [k(v) + 1]$, we slightly abuse notation and use $\lambda(v, i)$ to refer to both the counter itself, and its value at the end of the algorithm.

**Proposition A.3** (Penalty Robustness). *For every* $(q, \pi) \in Q$, *it holds that* $\text{OA}(q, \pi) \leq 2\pi$.

*Proof.* If UPONREQUEST($q, \pi$) returns in Line 22, then it must be that the condition in Line 9 has failed, and thus $\tau + \delta(u, q) \leq \pi$; thus, $\text{OA}(q, \pi) = \tau + (\tau + \delta(u, q)) \leq 2\pi$.

Otherwise, UPONREQUEST($q, \pi$) returned on Line 11, in which case note that since $\tau$ is raised continuously from 0, Line 11 ensures that $\tau \leq \pi$ at all times. Thus, $\text{OA}(q, \pi) = \tau + \pi \leq 2\pi$, completing the proof. $\qquad\square$

It remains to show subset competitiveness for the algorithm. Henceforth, fix any subset of the input $Q' \subseteq Q$.

**Proposition A.4.** *For every request $q$ and $v \in T$, $\tau_q^v \leq c(v)$.*

*Proof.* Observe that $\tau_q^v$ cannot exceed $\delta_{t(v)}^v$, for some current value of $t(v)$, or else the request is connected (or escalated to a parent node). The fact that $\delta_0^v = c(v)$, together with the fact that $\delta_i^v$ is a decreasing sequence in $i$ (Observation 3.4) complete the proof. $\qquad\square$

We now begin to bound the (amortized) costs of the algorithm. Recall that $\overline{Q'}$ is the input $Q'$ with the penalties set to infinity; that is, the prize-collecting input converted to the standard setting. We would like to prove the following lemma.

**Lemma A.5.** $\mathbb{E}[\text{OA}(Q'|Q)] \leq O(\log(|S|)) \cdot \text{OPT}_S\left(\overline{Q'}\right)$.

When the input consists of requests that are also from $S$, both the clients and facilities are from $S$, and thus on the leaves of the tree $T$. In this case, we define $\text{OPT}_T$ to be any solution for the input under the metric space induced by the weighted HST $T$. To prove Lemma A.5, we first bound the cost of the algorithm against $\text{OPT}_T$ on a set of clients mapped to their closest neighbors in $S$.

**Lemma A.6.** *Let $Q'_S$ be the input formed from $Q'$ by mapping each request $(q, \pi) \in Q'$ to the request $(p(q), \pi)$. It holds that*

$$\text{OA}(Q'|Q) \leq \sum_{(q,\pi) \in Q'} \delta(q, S) + O(D) \cdot \text{OPT}_T^f\left(\overline{Q'_S}\right) + O(1) \cdot \text{OPT}_T^c\left(\overline{Q'_S}\right)$$

*Proof.* First, observe both **return** statements in Algorithm 2 and note that for every request $(q, \pi) \in Q$ it holds that

$$\text{OA}(q, \pi) \leq 2\tau_q + \delta(q, S). \tag{8}$$

We now focus on bounding $\sum_{(q,\pi) \in Q'} \tau_q$, i.e., total amount by which counters are raised when handling $Q'$. Let $w$ be a facility opened in $\text{OPT}_T(\overline{Q'_S})$. Let $R \subseteq Q'$ be the set of requests such that their corresponding requests in $\overline{Q'_S}$ are connected by $\text{OPT}_T$ to the facility $w$. Using Observation 3.4, for every ancestor tree node $v$ of $w$, we define $i_v$ to be the minimal index such that $\ell(o_{i_v}^v) \leq \ell(o_w)$ and $\ell(\delta_{i_v}^v) \leq \ell(\delta_T(v, w))$.

Let $P(w) = (v_0 = w, v_1, \cdots, v_k = r)$ be the path from $w$ to the root. The sum $\sum_{(q,\pi) \in Q'} \tau_q$ can be divided as follows:

1. Raising counters $\lambda(v, i)$ for $v \in P(w)$, $i \leq i_v$. The total amount here is at most

$$\sum_{v \in P(w)} \sum_{i=1}^{i_v} \lambda(v, i) \leq \sum_{v \in P(w)} \sum_{i=1}^{i_v} o_i^v \leq \sum_{i=1}^{i_v} 2^{\ell(o_i^v)+1} \leq \sum_{v \in P(w)} 2^{\ell(o_{i_v}^v)+2}$$

$$\leq \sum_{v \in P(w)} 2^{\ell(o_w)+2} \leq \sum_{v \in P(w)} 4o_w \leq 4Do_w.$$

2. Raising counters $\lambda(v, i)$ for $v \notin P(w)$. Consider a request $q \in R$, and define $u := p(q) = u$ and $v$ to be the lowest common ancestor of $u$ and $w$. The only nodes not in $P(w)$ in which counters are raised when handling $q$ are on the path from $u$ (inclusive) to $v$ (non-inclusive). Using Proposition A.4, the total increase in counters for these nodes is at most $\delta_T(u, v)$.

3. Raising counters $\lambda(v, i)$ for $v \in P(w)$ and $i > i_v$. Suppose that a request $q$ raises such a counter $\lambda(v_j, i)$ for some node $v_j \in P(w)$. When such a counter is raised, the proxy $s_{i_{v_j}}^{v_j}$ is already open, and thus the total raising of counters of index greater than $i_{v_j}$ for $v_j$ by $q$ is at most $\delta_{i_{v_j}}^{v_j} \leq 2\delta_w^{v_j} = 2\delta_T(v_j, v) + 2\delta_T(v, w)$, where $v$ is the lowest common ancestor of $u$ and $w$. (Note that other proxies of $v_j$ of larger index could be open, but they can only be closer to $v_j$, thus limiting the raising of counters even further.)

Of those two costs, we would like to charge $q$ only for $2\delta_T(v, w)$, and charge $2\delta_T(v_j, v)$ in aggregate over all $q$. To do so, observe that the counters for nodes in $P(w) \setminus \{v_j\}$ that were raised upon request $q$ must be of the form $\lambda(v_l, 1)$ for $v_l \in \{v_0, \cdots, v_{j-1}\}$. As the request $q$ was repeatedly escalated from $v$ to $v_j$, the total increase in those counters must be at least $\delta_T(v, v_j)$, and thus $2\delta_T(v, v_j)$ is upper bounded by twice the increase in those counters. However, as seen in Item 1, over all requests, these increases sum to at most $4Do_w$ over all $q \in R$; thus, the term $2\delta_T(v_j, v)$ sums in aggregate to at most $8Do_w$.

Overall, denoting by $w^q$ the lowest common ancestor of $p(q)$ and $w$, we get:

$$\sum_{(q,\pi) \in R} \tau(q, \pi) \leq 4Do_w + \sum_{(q,\pi) \in R} \delta_T(p(q), w^q) + \left( 8Do_w + \sum_{(q,\pi) \in R} \delta_T(w^q, w) \right)$$
$$\leq 12Do_w + 2\delta_T(p(q), w).$$

Summing over all $w$, we get

$$\sum_{(q,\pi) \in Q'} \tau(q, \pi) \leq 12D \cdot \mathrm{OPT}_T^f\left(\overline{Q_S'}\right) + 2 \cdot \mathrm{OPT}_T^c\left(\overline{Q_S'}\right).$$

Combining with Equation (8), we get

$$\mathrm{OA}(Q'|Q) \leq \sum_{(q,\pi) \in Q'} \delta(q, S) + 24D \cdot \mathrm{OPT}_T^f\left(\overline{Q_S'}\right) + 4 \cdot \mathrm{OPT}_T^c\left(\overline{Q_S'}\right). \qquad \square$$

Having bounded the costs of the algorithm against $\mathrm{OPT}_T$, we can now prove Lemma A.5.

*Proof of Lemma A.5.* Using Lemma A.6, we get the following.

$$\mathbb{E}[\mathrm{OA}(Q'|Q)] \leq \sum_{(q,\pi) \in Q'} \delta(q, S) + \mathbb{E}\left[ O(\log(|S|)) \cdot \mathrm{OPT}_T^f\left(\overline{Q_S'}\right) + O(1) \cdot \mathrm{OPT}_T^c\left(\overline{Q_S'}\right) \right]$$

Now, note that every solution $\mathrm{OPT}_S(\overline{Q_S'})$ induces a solution for $\overline{Q_S'}$ on $T$, which opens the same facilities and makes the same connections (through the tree); the new tree solution has the same facility opening costs, and connection costs which are, in expectation, at most $O(\log(|S|))$-times greater (see Theorem 3.3). Thus, we have

$$\mathbb{E}[\mathrm{OA}(Q'|Q)] \leq \sum_{(q,\pi) \in Q'} \delta(q, S) + O(\log(|S|)) \cdot \mathrm{OPT}_S\left(\overline{Q_S'}\right)$$

Now, note that any solution $\mathrm{OPT}_S\left(\overline{Q'}\right)$ induces a solution for $\overline{Q_S'}$ of cost $\sum_{(q,\pi) \in Q'} \delta(q, S) + \mathrm{OPT}_S\left(\overline{Q'}\right)$, and also note that $\sum_{(q,\pi) \in Q'} \delta(q, S)$ is a lower bound for $\mathrm{OPT}_S\left(\overline{Q'}\right)$. Plugging into the displayed equation above completes the proof of the lemma. $\qquad \square$

*Proof of Theorem 3.1.* Lemma A.1 and Observation A.2 show that the online amortization $\mathrm{OA}$ is valid and monotone. Proposition A.3 shows penalty robustness, while Lemma A.5 shows subset competitiveness; thus, the algorithm is $O(\log|S|)$-PRSC using $\mathrm{OA}$ w.r.t. $S$. Using Proposition C.2, the algorithm is Lagrangian $O(\log|S|)$-subset-competitive using $\mathrm{OA}$ w.r.t. $S$. $\qquad \square$

---
**Algorithm 4:** Variant of Fotakis' Algorithm for Prize-Collecting OFLP
---

1 **Initialization**
2   | Let $Q \leftarrow \emptyset$.
3   | Let $F \leftarrow \emptyset$.
4   | For every $v \in X$, let $p(v) \leftarrow 0$.

5 **Event Function** UPONREQUEST$(q, \pi)$ // *Upon the next request $q$ in the sequence on point $u \in X$*
6   | Set $Q \leftarrow Q \cup \{q\}$.
7   | Denote by $v_0$ the closest open facility to $q$.
8   | Define $\tau_q \leftarrow \min\{\pi, \delta(q, F), \min_{v \in X}\{o_v - p(v) + \delta(q, v)\}\}$
9   | **if** $\tau_q = \delta(q, F)$ **then**
10   |   | Connect $q$ to the closest facility in $F$.

11   | **else if** $\tau_q = o_v - p(v) + \delta(q, v)$ *for some $v \in X$* **then**
12   |   | Open a facility at $v$.
13   |   | Connect $q$ to $v$.

14   | **else**
15   |   | Pay the penalty $\pi$ for $q$.

16   | COMPUTEPOTENTIALS()
17   | **return** $2\tau_q$ // *return amortized cost*

18 **Function** COMPUTEPOTENTIALS()
19   | For every $q \in Q$, define $\lambda_q = \min\{\delta(q, F), \tau_q\}$
20   | For every location $v \in X$, set $p(v) \leftarrow \sum_{q \in Q}(\lambda_q - \delta(q, v))^+$.

# B   Online Facility Location: The $O(\log n)$-Competitive Algorithm

In this section, we present and analyze a prize-collecting algorithm for facility location with predictions whose competitive ratio on the number of requests $n = |Q|$. As is required for using Algorithm 1, this algorithm is Lagrangian subset-competitive. This algorithm is based on the work of Fotakis [18] for the non-prize-collecting setting. Specifically, we prove the following theorem.

**Theorem B.1.** *For facility location with predictions, there exists a deterministic prize-collecting algorithm* ALG *with a monotone online amortization* OA *which is Lagrangian $O(\log n)$-subset-competitive using* OA.

## B.1   The Algorithm

**Algorithm's description.** This algorithm follows the main principles of Fotakis [18]. Each point in the metric space has an associated potential, such that when that potential exceeds the cost of opening a facility at that point, the facility is opened. This potential roughly translates to the amount by which the cost of the offline solution for known requests would decrease by opening a facility at that location. Observing each request, consider the ball centered at that request such that the closest open facility lies on the sphere of that ball; the request imposes a potential increase for every point inside that ball. However, as the requests now have penalties, these penalties cap the radius of the ball, i.e., limit the potential imposed by the requests.

Specifically, the algorithm assigns each request a cost $\tau_q$, which intuitively is the minimum cost of handling the current request. This cost could be the penalty cost, the cost of connecting to an open facility, or the cost of opening a facility (beyond the current potential budget) and then connecting to it. The algorithm spends an amortized cost of $\tau_q$ to serve $q$, but a potential ball of radius $\tau_q$ is also created to serve future requests (at an future cost of at most $\tau_q$).

For every $x$, we use $x^+$ as a shorthand for $\max\{0, x\}$. The prize-collecting algorithm based on [18] is given in Algorithm 4.

### B.1.1   Analysis

We now analyze Algorithm 4 and show that it proves Theorem B.1. For this analysis, we fix the prize-collecting input $Q$. Next, we define the online amortization OA such that OA$(q, \pi)$ is the value returned by UPONREQUEST in Theorem B.1 upon release of $(q, \pi) \in Q$.

**Online Amortization**

We first prove that OA is valid and monotone.

**Lemma B.2.** *The online amortization* OA *for Algorithm 4 is valid, i.e.,* $\mathrm{ALG}(Q) \leq \mathrm{OA}(Q)$.

*Proof.* For each request, observe the variable $\tau_q$, and note that:

- If the penalty $\pi$ is paid for $q$, then $\tau_q = \pi$.

- If $q$ is connected to some facility, the connection cost of $q$ does not exceed $\tau_q$.

It remains to bound the opening costs of the algorithm. Observe the evolution of the potential function $\sum_{q \in Q} \min\{\delta(q, F), \tau_q\}$ as $Q$ and $F$ grow over time. This function is nonnegative, and grows by exactly $\tau_q$ upon the release of $(q, \pi)$ (after Line 8). Moreover, whenever a facility at $v$ is opened (thus joining $F$), it decreases this amount by exactly $o_v$. Thus, the total opening cost can be bounded by $\sum_{q \in Q} \tau_q$.

Overall, we bounded the cost of the algorithm by $\sum_{(q,\pi) \in Q} 2\tau_q = \sum_{(q,\pi) \in Q} \mathrm{OA}(q, \pi)$. $\square$

**Observation B.3.** *The online amortization* OA *given for Algorithm 4 is a monotone online amortization.*

## B.2 Bounding Amortized Costs

Having shown the necessary properties for the online amortization, we proceed to show that Algorithm 4 is Lagrangian subset-competitive using this amortization. As in Section 3, we first show that the algorithm is PRSC (see Proposition C.2); we begin by observing the penalty robustness of the algorithm.

**Observation B.4.** *For every* $(q, \pi) \in Q$, *it holds that* $\mathrm{OA}(q, \pi) \leq 2\pi$.

We now fix the subset $Q' \subseteq Q$ for the sake of proving subset competitiveness. Recall that $\overline{Q'}$ is the standard input formed from the prize-collecting input $Q'$ (by setting penalties to infinity).

Before proving subset-competitiveness, we need to prove the following simple lemma.

**Lemma B.5** (Min trace lemma). *Let* $(a_1, \cdots, a_k), (b_1, \cdots, b_k)$ *be two sequences of non-negative numbers, and define* $c_{i,j} = \min(a_i, b_j)$. *Then if there exists* $z$ *such that for every* $i$ *it holds that* $\sum_{j=1}^{i} c_{i,j} \leq z$, *then it holds that* $\sum_{i=1}^{k} c_{i,i} = O(\log k) \cdot z$.

*Proof.* We prove that $\sum_{i=1}^{k} c_{i,i} \leq H_k \cdot z$ by induction on $k$, where $H_k = \sum_{i=1}^{k} \frac{1}{i}$ is the $k$-th harmonic number. Note that the base case, in which $k = 1$, holds as $c_{1,1} \leq z$.

Now, for the general case, note that if we can find $i$ such that $c_{i,i} \leq \frac{z}{k}$, then we can complete the proof by induction on the sequences $(a_1, \cdots, a_{i-1}, a_{i+1}, \cdots, a_k)$ and $(b_1, \cdots, b_{i-1}, b_{i+1}, \cdots, b_k)$. (Note that the constraints required for this inductive instance are implied by the original constraints.) This induction would imply that $\sum_{i' \neq i} c_{i',i'} \leq H_{k-1} \cdot z$, to which adding $c_{i,i}$ would complete the proof.

It remains to find $i$ such $c_{i,i} \leq \frac{z}{k}$. We consider the constraint $\sum_{j=1}^{k} c_{k,j} \leq z$, and observe the following cases.

**Case 1**: $c_{k,j}$ are equal for all $j$. In this case, all $c_{k,j}$ are at most $\frac{z}{k}$. In particular, this is true for $c_{k,k}$; thus, choosing $i = k$ completes the proof.

**Case 2**: $c_{k,j}$ are not all equal. In this case, observe $j$ that minimizes $c_{k,j}$, and note that $c_{k,j} \leq \frac{z}{k}$. There exists $j'$ such that $c_{k,j} < c_{k,j'}$, which implies $c_{k,j} < a_k$, and thus $c_{k,j} = b_j$, yielding $b_j \leq \frac{z}{k}$. But this implies $c_{j,j} \leq b_j \leq \frac{z}{k}$, and thus choosing $i = j$ completes the proof. $\square$

We can now prove subset-competitiveness, as stated in Lemma B.6.

**Lemma B.6.** $\mathrm{OA}(Q'|Q) \leq O(\log|Q'|) \cdot \mathrm{OPT}(\overline{Q'})$.

*Proof.* Let $w$ be some facility opened by $\text{OPT}\left(\overline{Q'}\right)$, and denote by $R \subseteq \overline{Q'}$ the set of requests connected to that facility in $\text{OPT}\left(\overline{Q'}\right)$. Define $C_w := \sum_{(q,\pi) \in R} \delta(w, q)$ the total connection cost incurred by $\text{OPT}\left(\overline{Q'}\right)$ on the facility $w$. Enumerate these requests as $((q_1, \pi_1), \cdots, (q_k, \pi_k))$, where $k = |R|$. For $1 \leq i \leq k$, denote by $F_i$ the set of facilities which were open immediately before the release of $(q_i, \pi_i)$. As a shorthand, we also define $\tau_i = \tau_{q_i}$. Consider that the total potential of the facility $w$ can never exceed its cost $o_w$; moreover, upon release of $(q_i, \pi_i)$, the choice of $\tau_i$ ensures that

$$o_w \geq \tau_i - \delta(q_i, w) + \sum_{j=1}^{i-1} \min(\tau_j, (\delta(q_j, F_i) - \delta(q_j, w))^+)$$

$$\geq \tau_i - \delta(q_i, w) + \sum_{j=1}^{i-1} \min(\tau_j, \delta(q_i, F_i) - \delta(q_i, w) - \delta(q_j, w) - \delta(q_j, w))$$

$$\geq \tau_i - \delta(q_i, w) + \sum_{j=1}^{i-1} \min(\tau_j, \tau_i - \delta(q_i, w)) - 2\sum_{j=1}^{i-1} \delta(q_j, w)$$

$$\geq \tau_i - \delta(q_i, w) + \sum_{j=1}^{i-1} \min(\tau_j, \tau_i - \delta(q_i, w)) - 2C_w$$

$$\geq \sum_{j=1}^{i} \min(\tau_j, \tau_i - \delta(q_i, w)) - 2C_w \tag{9}$$

where the second inequality uses the triangle inequality and the third inequality uses the definition of $\tau_i$.

From Equation (9), we have that for every $1 \leq i \leq k$ it holds that

$$\sum_{j=1}^{i} \min(\tau_j, \tau_i - \delta(q_i, w)) \leq o_w + 2C_w.$$

Using Lemma B.5, this yields

$$\sum_{i=1}^{k} \min(\tau_i, \tau_i - \delta(q_i, w)) \leq O(\log k) \cdot (o_w + 2C_w)$$

Since $\tau_i - \delta(q_i, w) = \min(\tau_i, \tau_i - \delta(q_i, w))$, and since $\sum_{i=1}^{k} \delta(q_i, w) = C_w$, we have

$$\sum_{i=1}^{k} \tau_i \leq O(\log k) \cdot (o_w + C_w) \leq O(\log|Q'|) \cdot (o_w + C_w)$$

Finally, summing over all facilities $w$ in $\text{OPT}(\overline{Q'})$ yields

$$\sum_{(q,\pi) \in Q'} \tau_q \leq O(\log|Q'|)\text{OPT}(\overline{Q'}). \qquad \square$$

*Proof of Theorem B.1.* Through Lemma B.2 and Observation B.3, we have that OA is a valid and monotone amortization for Algorithm 4. Lemma B.6 and Observation B.4 then yield that the algorithm is $O(\log Q)$-PRSC using OA. Using Proposition C.2 yields that the algorithm is Lagrangian $O(\log Q)$-subset-competitive using OA, which completes the proof of the theorem. $\qquad \square$

## C  The Smooth Combination Framework

### C.1  Proof of Theorem 2.5

*Proof of Theorem 2.5.* Consider the framework in Algorithm 1 applied to algorithms $\text{ALG}_1$, $\text{ALG}_2$. The framework ensures that all requests are satisfied, as at least one of the constituent algorithms

serves each request. Denote by $\alpha(q)$ the final value assigned to the variable $\alpha$ upon request $q$; the prize-collecting input given to both constituent algorithms is $Q^* = ((q, \alpha(q)))_{q \in Q}$. We define $Q_1^*, Q_2^*$ be the partition of $Q^*$ induced by the partition of $Q$ into $Q_1, Q_2$. As the algorithm only buys items bought by one of the constituent algorithms, its cost can thus be bounded by $\text{ALG}_1(Q^*) + \text{ALG}_2(Q^*)$. We now bound $\text{ALG}_1(Q^*)$; bounding $\text{ALG}_2(Q^*)$ is identical.

First, consider the prize-collecting solution which serves $Q_1^*$ optimally subject to using items from $S$, but pays the penalty for requests from $Q_2^*$; using the Lagrangian subset-competitiveness of $\text{ALG}_1$ against this solution yields

$$\mathbb{E}[\text{ALG}_1(Q^*)] \leq O(\beta_1) \cdot \text{OPT}_S(Q_1) + \mathbb{E}\left[O(1) \cdot \sum_{q \in Q_2} \alpha(q)\right] \tag{10}$$

Now, observe that using the definition of $\alpha$ and the fact that $\text{ALG}_2$ is monotone, we have that $\alpha(q) \leq \text{ALG}_2(q, \alpha(q))$; summing over requests in $Q_2$ we get that $\sum_{q \in Q_2} \alpha(q) \leq \text{ALG}_2(Q_2^*|Q^*))$. Plugging into Equation (10), we get

$$\mathbb{E}[\text{ALG}_1(Q^*)] \leq O(\beta_1) \cdot \text{OPT}_S(Q_1) + \mathbb{E}\left[O(1) \cdot \text{ALG}_2(Q_2^*|Q^*))\right]$$
$$\leq O(\beta_1) \cdot \text{OPT}_S(Q_1) + O(\beta_2) \cdot \text{OPT}(Q_2)$$

where the second inequality uses the fact that $\text{ALG}_2$ is subset competitive to bound its cost on the subset $Q_2^*$ against the solution which serves those requests optimally. This completes the bounding of costs for $\text{ALG}_1$; we can bound $\mathbb{E}[\text{ALG}_2(Q^*)]$ in the same way. Summing the bounds for $\text{ALG}_1$ and $\text{ALG}_2$, we get

$$\text{ALG}(Q) \leq O(\beta_1) \cdot \text{OPT}_S(Q_1) + O(\beta_2) \cdot \text{OPT}(Q_2)$$

which completes the proof. $\qquad\square$

## C.2 Penalty-Robust Subset-Competitive Algorithms

In proving that a prize-collecting algorithm is Lagrangian subset-competitive (for use in our framework), we sometimes find it easier to prove that it is *penalty-robust subset competitive*. As we now prove, this latter property is sufficient to prove the former. (In fact, it is easy to see that both properties are in fact equivalent.)

**Definition C.1** (PRSC algorithm using online amortization). Let ALG be a randomized prize-collecting algorithm equipped with an online amortization OA running on an input $Q$. We say that ALG is $\beta$ penalty-robust subset competitive (PRSC) using OA if both following conditions hold:

1. For every $(q, \pi) \in Q$ we have $\text{OA}(q, \pi) \leq O(1) \cdot \pi$.

2. For every subset $Q' \subseteq Q$, we have $\mathbb{E}[\text{OA}(Q'|Q)] \leq \beta \cdot \text{OPT}(\overline{Q'})$.
   (where $\overline{Q'}$ is the input formed from $Q'$ by forcing service, i.e., setting penalties to infinity.)

If in the second condition of PRSC we replace $\text{OPT}\left(\overline{Q'}\right)$ by $\text{OPT}_S\left(\overline{Q'}\right)$, we say that ALG is $\beta$-PRSC using OA w.r.t. $S$.

**Proposition C.2.** *A $\beta$-PRSC algorithm using OA (w.r.t. S) is also Lagrangian $\beta$-subset-competitive using OA (w.r.t. S).*

*Proof.* We prove this for a general solution, restricting to $S$ is identical. Consider prize-collecting input $Q$, and any subset $Q' \subseteq Q$. Let SOL be the optimal solution for $Q'$, which pays penalties for $Q'_p$ and serves $Q'_b = Q' \setminus Q'_p$ optimally. Then it holds that

$$\mathbb{E}[\text{OA}(Q'|Q)] = \mathbb{E}\left[\text{OA}(Q'_b|Q)\right] + \mathbb{E}\left[\text{OA}\left(Q'_p|Q\right)\right]$$
$$\leq \beta \cdot \text{OPT}\left(\overline{Q'_b}\right) + O(1) \cdot \sum_{(q, \pi) \in Q'_p} \pi$$
$$= \beta \cdot \text{SOL}^b(Q') + O(1) \cdot \text{SOL}^p(Q')$$

where the inequality uses both properties of PRSC. $\qquad\square$

# D  Proofs of Theorems 1.1, 1.2 and 1.3

We establish these theorems in three steps. First, we combine various constituent prize-collecting algorithms using Theorem 2.5 and explicitly state the guarantees for the resulting algorithms. Then, we use these guarantees to derive the discrete-smoothness property for the individual problems with respect to the size of the prediction (i.e., Equation (6)). Finally, we use Theorem 2.1 to make the competitive ratio depend on $|S\backslash\text{OPT}|$ rather than on $|S|$.

Before proceeding further, we need to precisely define the intersection/difference of a solution with a prediction to make Theorem 1.1, Theorem 1.2, and Theorem 1.3 completely formal.

**Definition D.1** (restriction of solution with prediction)**.** Consider an online covering problem with items $\mathcal{E}$, let $S \subseteq \mathcal{E}$ be some prediction. For every solution $A$ which buys some items from $\mathcal{E}$:

- Define $A|_S$ to be the solution which only buys items from $S$, to the same amount as $A$.

- Define $A|_{\overline{S}}$ to be the solution which only buys items outside $S$, to the same amount as $A$.

**Facility Location with Predictions.** In order to describe facility location as a covering problem, we must describe the set of items. Here, the set of items comprises an opening item $b_v$ for each facility and a connection item $c_{v,q}$ for each (request, facility) pair. When we informally write that $S$ is a set of possible facilities, this can be formalized to the set of items $b_v$ for $v \in S$, plus the connection items $c_{v,q}$ for all $q$ in the input and $v \in S$.

Due to Theorem 3.1 and Theorem B.1, we have that both Algorithm 2 and Algorithm 4 can serve as constituent algorithms in our framework. Combining both algorithms using Theorem 2.5 thus implies the following theorem.

**Theorem D.2.** *For facility location with predictions, there exists a randomized algorithm* ALG *such that for every input $Q$, and for every partition of $Q$ into $Q_1, Q_2$, we have*

$$\mathbb{E}[\text{ALG}(Q)] \leq O(\log|S\backslash\text{OPT}|) \cdot \text{OPT}_S(Q_1) + O(\log|Q_2|) \cdot \text{OPT}(Q_2).$$

We obtain an additional result, which is useful for small metric spaces, from combining two instances of Algorithm 2, one for the entire metric space $X$ and one for the predictions $S$.

**Theorem D.3.** *For facility location with predictions, there exists a randomized algorithm* ALG *such that for every input $Q$, and for every partition of $Q$ into $Q_1, Q_2$, we have*

$$\mathbb{E}[\text{ALG}(Q)] \leq O(\log|S\backslash\text{OPT}|) \cdot \text{OPT}_S(Q_1) + O(\log|X|) \cdot \text{OPT}(Q_2).$$

*Proof of Theorem 1.1.* Consider a solution OPT to facility location on a set of requests $Q$. Partition $Q$ into $Q_1, Q_2$ such that $Q_1$ contains all requests from $Q$ that are connected to a facility in $\text{OPT}|_S$ (and $Q_2$ is complementary). Using the algorithm ALG from Theorem D.2, we have

$$\text{ALG}(Q) \leq O(\log|S|) \cdot \text{OPT}_S(Q_1) + O(\log|Q_2|) \cdot \text{OPT}(Q_2). \tag{11}$$

Now note that $\text{OPT}|_S$ is a solution to $Q_1$ that only uses facility and connection items from $S$, and thus $\text{OPT}_S(Q_1) \leq \text{OPT}|_S$. Moreover, $\text{OPT}|_{\overline{S}}$ is a solution to $Q_2$, and thus $\text{OPT}_S(Q_2) \leq \text{OPT}|_{\overline{S}}$. Plugging into Equation (12), and noting that $|Q_2| \leq |Q|$, we get

$$\text{ALG}(Q) \leq O(\log|S|) \cdot \text{OPT}|_S + O(\log|Q|) \cdot \text{OPT}|_{\overline{S}}.$$

We now plug the above equation into Theorem 2.1, thus replacing the dependence on $|S|$ with dependence on $|S\backslash\text{OPT}|$. $\qquad\square$

*Proof of Theorem 1.2.* Identical to the proof of Theorem 1.1, but using Theorem D.3. $\qquad\square$

**Set Cover with Predictions.** Theorem 4.1 implies that Algorithm 3 is Lagrangian subset-competitive. In addition, it is easy to see that Algorithm 3 is monotone, as defined in Definition 2.2. Thus, the algorithm can serve as a constituent algorithm in our framework. From combining two instances of Algorithm 3, Theorem 2.5 thus implies the following theorem.

**Theorem D.4.** *For fractional set cover with predictions, with universe $(E, U)$ and a prediction $S \subseteq U$, there exists a deterministic algorithm* ALG *such that for every input $Q$, and for every partition of $Q$ into $Q_1, Q_2$, we have*

$$\text{ALG}(Q) \leq O(\log|S|) \cdot \text{OPT}_S(Q_1) + O(\log|U|) \cdot \text{OPT}(Q_2).$$

Using standard rounding techniques (see [3, 15]) for online set cover, we can round the fractional solution online at a loss of $O(\log|Q|)$. In addition, we can then apply Theorem 2.1 to replace $|S|$ with $|S \backslash \mathrm{OPT}|$. Thus, Theorem D.4 yields the following corollary.

**Corollary D.5.** *For (integral) set cover with predictions, with universe $(E, U)$ and a prediction $S \subseteq U$, there exists a randomized algorithm* ALG *such that for every input $Q$, and for every partition of $Q$ into $Q_1, Q_2$, we have*

$$\mathbb{E}[\mathrm{ALG}(Q)] \leq O(\log|Q| \log|S \backslash \mathrm{OPT}|) \cdot \mathrm{OPT}_S(Q_1) + O(\log|Q| \log|U|) \cdot \mathrm{OPT}(Q_2).$$

*Proof of Theorem 1.3.* Consider a solution OPT to set cover on a set of requests $Q$. Partition $Q$ into $Q_1, Q_2$ such that $Q_1$ contains all requests from $Q$ that belong to a set in $\mathrm{OPT}|_S$ (and $Q_2$ is complementary). Using the randomized algorithm ALG from Corollary D.5, we have

$$\mathrm{ALG}(Q) \leq O(\log|Q| \log|S|) \cdot \mathrm{OPT}_S(Q_1) + O(\log|Q| \log|U|) \cdot \mathrm{OPT}(Q_2). \tag{12}$$

Now note that $\mathrm{OPT}|_S$ is a solution to $Q_1$ that only uses sets from $S$, and thus $\mathrm{OPT}_S(Q_1) \leq \mathrm{OPT}|_S$. Moreover, $\mathrm{OPT}|_{\overline{S}}$ is a solution to $Q_2$, and thus $\mathrm{OPT}_S(Q_2) \leq \mathrm{OPT}|_{\overline{S}}$. Plugging into Equation (12), we get

$$\mathrm{ALG}(Q) \leq O(\log|Q| \log|S|) \cdot \mathrm{OPT}|_S + O(\log|Q| \log|U|) \cdot \mathrm{OPT}|_{\overline{S}}.$$

$\square$

# E   Proof of Theorem 2.1: Reduction from Equation (6) to Equation (1)

In this section, we give the proof of Theorem 2.1 whose goal is to give a reduction from Equation (6) to Equation (1). This replaces $s$ in the bound of Equation (6) with the term $s_\delta$, where $s_\delta := |S \backslash \mathrm{OPT}|$, in order to obtain Equation (1).

*Proof of Theorem 2.1.* Assume, without loss of generality, that the cheapest item in $\mathcal{E}$ costs 1. Consider the following construction of the algorithm ALG using the algorithm ALG':

---
1 Initialize $i \leftarrow 0$, $S' \leftarrow S$, $B \leftarrow 0$, and define the item cost function $c' \leftarrow c$.
2 Let $A$ be an instance of ALG' with prediction set $S'$, and cost function $c'$.
3 **for** *incoming request $q$* **do**
4   **while** *True* **do**
5     Simulate sending $q$ to $A$, and let $c$ be the resulting cost.
6     **if** $B + c < 2^i$ **then break**
7     Spend $2^i$ budget in buying the cheapest items in $S'$, let the bought subset of items be $T$.
8     Set $S' \leftarrow S' \backslash T$, $B \leftarrow 0$, $i \leftarrow i + 1$.
9     For every $e \in T$, set $c'(e) \leftarrow 0$.
10     Reset $A$ to be a new instance of ALG', given $S'$ as prediction, and using the (modified) cost function $c'$.
11   Send $q$ to $A$, and set $B \leftarrow B + c$.
---

For integer $\ell$, define *phase $\ell$* to be the subsequence of requests in which variable $i$ takes value $\ell$. The cost of the algorithm can be charged to a constant times $2^j$, where $j$ is the penultimate value of $i$ in the algorithm. If $2^{j-1} < \mathrm{OPT}$, then the cost of the algorithm is at most $O(1) \cdot \mathrm{OPT}$ and we are done. Henceforth, suppose $\mathrm{OPT} \leq 2^{j-1}$. Define $S'_j, A_j, c'_j$ to be the values of the variables $S'$, $A$ and $c'$ during phase $j$. When considering the cost of a solution relative to a cost function, we place that cost function as superscript (e.g., $\mathrm{OPT}^{c'_j}$). Before the beginning of phase $j$, the algorithm spent at least OPT budget on buying the cheapest items in the (remaining) prediction; it thus holds that $\left|S'_j\right| \leq |S \backslash \mathrm{OPT}|$. Let $q_1, \cdots, q_k$ be the requests of phase $j$; moreover, let $q_{k+1}$ be the request upon which the variable $i$ was incremented to $j + 1$. From the definition of $q_{k+1}$, it holds that the cost of the instance of $A$ in phase $j$ on $(q_1, \cdots, q_k, q_{k+1})$ is at least $2^j$; thus, the total cost of the algorithm can be charged to this cost, which we denote by $\alpha$. But, through Equation (6), and from the fact that

OPT is a solution which serves $(q_1, \cdots, q_{k+1})$, we have

$$\alpha \leq O(f(|S \backslash \text{OPT}|)) \cdot \text{OPT}^{c'_j}|_{S'_j} + O(g) \cdot \text{OPT}^{c'_j}|_{\overline{S'_j}}$$

$$\leq O(f(|S \backslash \text{OPT}|)) \cdot \text{OPT}|_{S'_j} + O(g) \cdot \left( \text{OPT}^{c'_j}|_{\overline{S}} + \text{OPT}^{c'_j}|_{S \backslash S'_j} \right)$$

$$\leq O(f(|S \backslash \text{OPT}|)) \cdot \text{OPT}|_S + O(g) \cdot \text{OPT}|_{\overline{S}} \qquad \square$$

# F   Proof of Lemma 4.4

*Proof of Lemma 4.4.* First, note that $\text{ALG}(q, \pi) \leq 3 y_q$, where $y_q$ is the final value of the variable of that name: Proposition 4.5 implies that the buying cost is at most $2 y_q$, while a penalty of $\pi$ is paid only if $\pi \leq y_q$. We show that $\sum_{(q,\pi) \in Q'} y_q \leq O(\log m) \cdot \text{OPT}(\overline{Q'})$; since $\text{ALG}(q, \pi) \leq 3 \cdot y_q$, this would complete the proof of the lemma. Consider the (standard) primal and dual LPs for fractional set cover of $Q'$ without penalties (i.e. solving $\overline{Q'}$). The primal LP is given by:

$$\min \sum_{s \in U} x_s \cdot c(s) \text{ such that } \forall q \in Q' : \sum_{s | q \in s} x_s \geq 1 \text{ and } \forall s \in U : x_s \geq 0.$$

and the dual LP is given by:

$$\max \sum_{q \in Q'} y_q \text{ such that } \forall s \in U : \sum_{q | q \in s} y_q \leq c(s) \text{ and } \forall q \in Q' : y_q \geq 0.$$

We claim that the dual solution $\{y_q\}_{q \in Q'}$ violates dual constraints by at most $O(\log m)$; thus, scaling it down by that factor yields a feasible dual solution, and a lower bound to $\text{OPT}\left(\overline{Q'}\right)$.

Consider the dual constraint corresponding to the set $s$; we want to bound the term $\sum_{q \in Q' | q \in s} y_q$. Through induction on $k$, we can prove that once $\sum_{q \in Q' | q \in s} y_q = k$ for some integer $k$, it holds that $x_s \geq \frac{1}{m}\left( \left(1 + \frac{1}{c_s}\right)^k - 1 \right)$. Thus, once $k = \Theta(c_s \log m)$ we have $x_s \geq 1$, and $\sum_{q \in s} y_q$ would increase no more. This implies that scaling down $\{y_q\}_{q \in Q'}$ by $\Theta(\log m)$ yields a feasible dual solution, which lower bounds $\text{OPT}(\overline{Q'})$, and completes the proof.

It remains to prove the inductive claim. For the base case where $k = 0$, the claim holds trivially. Now, assume that the claim holds for $k - 1$, and consider point in which $\sum_{q \in Q' | q \in s} y_q$ is incremented from $k - 1$ to $k$; let $x, x'$ be the old and new amounts by which $s$ is held in the algorithm. We have

$$x' = x \cdot \left(1 + \frac{1}{c(s)}\right) + \frac{1}{U(q)c(s)} \geq \frac{1}{m}\left( \left(1 + \frac{1}{c(s)}\right)^k - 1 - \frac{1}{c(s)} \right) + \frac{1}{mc(s)} \geq \frac{1}{m}\left( \left(1 + \frac{1}{c(s)}\right)^k - 1 \right) \tag{13}$$

where the inequality uses the inductive hypothesis as well as the fact that $|U(q)| \leq m$. $\qquad \square$

