# OpenReview forum: "Discrete-Smoothness in Online Algorithms with Predictions"
_NeurIPS.cc/2023/Conference — NeurIPS 2023 poster_

### Official Review · Reviewer_mT48 · 2023-06-25

**Soundness:** 3 good
**Presentation:** 3 good
**Contribution:** 2 fair
**Rating:** 5
**Confidence:** 3

**Summary:**

This paper studies online algorithms with predictions, in which a prediction is provided to the algorithm without any guarantee on its quality. On this topic, the general goal is to provide an algorithm which is (1) consistent (i.e. if the prediction is perfect the algorithm should be much better than a pure online algorithm), (2) robust (i.e. maintain a worst-case guarantee even if the prediction is arbitrarily bad), and (3) interpolate smoothly between the cases (1) and (2) when the prediction is neither perfect nor horrible. To obtain (3) we need to define an error metric to quantify how good the prediction is.

In this paper, the authors propose a general property that they call “discrete-smoothness” and gives discrete-smooths algorithms for online set cover with predictions and online facility location with predictions. This discrete-smooth property can be seen as a general error-metric for combinatorial problems in which the prediction is given in the form of a predicted solution.

**Strengths:**

It is nice to have one general definition of smoothness for many problems (although this definition has some issues, see below). This is an interesting conceptual contribution.

**Weaknesses:**

The techniques do not seem very novel. The combiner algorithm (Algorithm 1) is a variant of an algorithm that has been rediscovered repeatedly in the recent years in the context of algorithm with predictions. The prize-collecting set cover seems like an easy adaptation of the classical online set cover. Algorithm 2 seems a bit more technical but I am not sure the techniques are very novel either ([19] already uses HSTs for this problem as far as I know).

It is nice to remark that one can get a quite general smoothness like this but to me this smoothness seems a bit like avoiding the difficulty. In my opinion, a very interesting part of algorithms with predictions is to come up with a natural error-metric which is easy to understand and to relate the performance of the algorithm to this metric. With this new definition, we are left with Equation (1) which is clearly not so simple. Also, discrete-smoothness is indeed general but it requires the prediction to be a whole solution. This is kind of the most elaborate prediction one can make and is probably not learnable for many problems.

**Questions:**

Line 86: I am not sure about the meaning of this comment “even for the online … (without predictions)….”. I thought Meyerson’s algorithm applied to the non-uniform case. Why are we comparing to Almanza in the case without predictions?

**Limitations:**

yes

---

> ### Author Rebuttal · Authors · 2023-08-09
>
> We thank the reviewer for their comments. Here is our response:
>
> * Regarding the novelty of the combiner, existing combiners do not achieve our smoothness guarantee. The standard online combiner achieves a competitiveness bound of $O(min(ALG_1(Q), ALG_2(Q)))$, which could be arbitrarily worse than our bound.
> To illustrate this, consider an input $Q$ in which ALG_1 suffers a cost of 1 on even-index requests and 1000 on odd-index requests, and ALG_2 suffers a cost of 1000 on even-index requests and 1 on odd-index requests. Both constituent algorithms would have a high cost on the input, of cost ~500 per request. This would also be the cost of a standard online combiner. However, our smooth combiner in Algorithm 1 would suffer a cost of ~1 per request.
>
> * Regarding the applicability of the discrete-smoothness notion, as mentioned in our general response, note that:
> Discrete-smoothness has meaning regardless of the manner in which the prediction is delivered to the algorithm. For example, for set cover, the prediction could offer a “recommended” set for every arriving element; the family of sets offered over the input would then be the prediction $S$.
>
> * The prediction could be any collection of items; in particular, we do not assume that the prediction is a feasible solution to the online input.
> Regarding online facility location without predictions, note that $m$ in our bound is the number of points in the metric space, rather than the number of requests; thus, we compare ourselves to Almanza et al. who gave a guarantee in this parameter. Indeed, in their paper they show that despite being log-competitive in the number of requests $n$, Meyerson’s algorithm is as bad as $\Omega(m)$-competitive.

---

> > ### Comment · Reviewer_mT48 · 2023-08-15
> >
> > Thank you for your response, and your clarifications.

---

### Official Review · Reviewer_h71T · 2023-06-28

**Soundness:** 3 good
**Presentation:** 3 good
**Contribution:** 3 good
**Rating:** 7
**Confidence:** 4

**Summary:**

This paper studies online facility location and online set cover with predictions, and provides improved theoretical guarantees for these problems. The authors state that these improvements come from a new property that they call discrete-smoothness. Learning-augmented algorithms take as input a prediction on the input, and then use this prediction to make better decisions. For problems considered in this paper, it makes sense to think about the projection of a prediction onto an optimal solution, so one can decompose a prediction into the part whose projection hits a fixed optimal solution and the part whose projection does not. In my own words, that’s what the discrete-smoothness property is capturing.

Technically, the authors introduce a property they call Lagrangian subset-competitiveness, which is a property on randomized prize collecting algorithms that bounds the algorithms performance on all subsets of the input. The fact that facility location and set cover both satisfy this property is what unifies them to be studied in this paper, and then thrown into the discrete-smoothness framework.

**Strengths:**

- Paper makes improvements on two learning-augmented problems. It is also worth noting that the assumptions on the settings are weaker in their work than in previous works.  In a sense, the results that existed for learning-augmented online set cover and facility location were a bit unsatisfying compared to what one should expect, so now the results seem up to par with what I would expect.
- I found the Lagrangian subset-competitiveness very interesting technically. Bounding an algorithm’s performance on all subsets is quite strong and it’s cool the authors were able to make use of this property
- Really clean writing.

**Weaknesses:**

- I don’t find the discrete-smoothness property technically impactful. The authors state it might be helpful as a general framework, but I think really what they found is that this Lagrangian subset-competitiveness property could be generally more useful. Do the authors have something in mind for when the discrete-smoothness property would be useful if an algorithm/ a problem does not satisfy Lagrangian subset-competitiveness? The only reason the discrete-smoothness property comes across as a weakness is because it’s highlighted a lot in the first 2 pages, so I was not really interested in the results until the third page.

**Questions:**

- (Please see question in Weaknesses, too)
- Some of the language used is a bit outdated (or not in keeping with the majority) for learning-augmented algorithms. For instance, your use of “robustness” is what the community refers to as “competitive” and your use of “smoothness” is what the community refers to as robustness. I know you’re following the style of some previous papers (like the Bamas et al work), but the community has converged to the 3 terms of competitive, robust, and consistent.
- The use of f() for the definition of discrete-smoothness and then f_v for the cost of opening a facility was a bit confusing, consider changing the variable of one?
- Did any of the previous work in learning-augmented facility location look at non-uniform opening costs, or were you the first?
- I’d say your statement that the Bamas et al work doesn’t satisfy any robustness property (in your definition of robustness) isn’t totally true. That work is interpolating between an offline and online algorithm based on a trust parameter, so in a sense the smoothness is in the trust parameter, which is some different type of measure of the error of the prediction.
- ^ In fact, from this view, part of what I found satisfying about your work is that now you smoothness is parameterized by a much better quantity.
- Your goal of hoping to define some more general, unifying framework for predictions is noble, but difficult. One line of work you might want to cite that has made some nice progress on this is by Sakaue and Oki. Check out their work on warm starts, if you aren’t familiar.
- I might be dense here, but part of why the discrete-smoothness property doesn’t stick with me is because I don’t understand how you think about the projection when the optimal solution isn’t unique… any thoughts here?

---

> ### Author Rebuttal · Authors · 2023-08-09
>
> We thank the reviewer for their comments. Regarding the more semantic comments, we will be sure to implement the necessary changes. Here are our responses to the remaining questions:
>
> * Regarding the question in the Weaknesses section, the discrete-smoothness guarantee makes sense for many problems; see, e.g., our general rebuttal which demonstrates the scope to which this guarantee is useful. The concept of Lagrangian subset-competitive prize-collecting algorithms (and combining them) was introduced for the sake of obtaining discrete-smoothness for facility location and set cover; however, for other problems (e.g., matching) achieving discrete smoothness could require different techniques. (However, it is worth mentioning that Lagrangian subset-competitive algorithms also yield some smoothness property for general covering problems, beyond facility location and set cover; see Theorem C.5 in the supplementary full paper).
>
> * Some previous works on non-uniform facility location with predictions are those of Jiang et al. [22] and Azar et al. [9]. Those works consider different prediction models than those considered in our paper.
>
> * Regarding the Bamas et al paper, thanks for the comment, we’ll incorporate it into the paper.
>
> * Regarding the non-uniqueness of the optimal solution, as mentioned in the paragraph starting at Line 52, our bound applies for every choice of a feasible solution OPT.
> To demonstrate why this is crucial, consider the case in which every item has two identical copies; in particular, this admits two optimal solutions OPT’,OPT’’ using the first and second sets of copies, respectively. Suppose the prediction is S=OPT’. Since OPT’’ is wholly unpredicted, If we insist on using OPT’’ in Equation (1), the bound becomes meaningless while the prediction itself is perfect.

---

> > ### Comment · Reviewer_h71T · 2023-08-16
> >
> > Thanks for the comments to my review.

---

### Official Review · Reviewer_9LeQ · 2023-07-04

**Soundness:** 2 fair
**Presentation:** 2 fair
**Contribution:** 2 fair
**Rating:** 3
**Confidence:** 3

**Summary:**

Ideal learning-augment online algorithms should have the properties of \emph{consistency} (meaning that they perform comparable to the optimum when the predictions are perfect) and \emph{robustness} (meaning that they don't perform much worse than their classic counterparts even for arbitrary predictions) and should interpolate between these two regimes smoothly as the quality of the predictions vary (\emph{smoothness}). The paper introduces a new form of smoothness guarantee called \emph{discrete-smoothness} and achieve discrete-smooth algorithms for online facility location and set cover improving over previous work for these two problems. The definition of discrete smoothness is quite opaque to me but if the predictor predicts a solution $S$ we would like the algorithm to achieve a value of at most $O(f(s_\Delta))\cdot OPT(S) +O(f(n))OPT(\overline{S})$ where  $OPT(S)=OPT\cap S$, $OPT(\overline{S})=OPT\setminus S$, $s_{\Delta}=|S\cap OPT|$, and $f$ is the competitive ration without predictions. I can see how this makes sense if $S$ is, e.g., a \emph{set} of points like for set cover or facility location.

The paper achieves these goals by designing algorithms for the prize-collecting variants of the problems (where the algorithms can choose to not include a vertex in set cover, or not serve a client in facility location at a penalty). It then combines these two algorithms by picking the minimum penalty such that the request is being served in one of the two algorithms. It finally 'buys' all items bought by either of the two algorithms. The first algorithm should have a better competitive ration of $O(f(s))$ but only on the restricted prediction $S$, the second one should have a competitive ratio of $O(f(n))$ 'against the unconstrained optimum OPT' which I am not sure what means.

**Strengths:**

It is very interesting to obtain a better understanding of the consistency-robustness tradeoff as the quality of the predictions varies in learning augmented algorithms, so the paper is well-motivated. The definition of discrete-smoothness could have some merit.

**Weaknesses:**

First of all, the paper is quite poorly written and it is hard even to make sense of the definition of discrete-smoothness. I can see what the definition means for set cover and facility location. However, as a general definition for combinatorial problems, I don't think that it is that sensible. $OPT|_{S}$ appear to be both a number (the cost of the solution restricted to $S$) but it is also a set of points $OPT\cap S$ (which by the way only seems to make sense in problems where the solution and prediction is a set of points). Even if the solutions are described as a set of points, is it necessarily the case that the cost is $|OPT\cap S|$ (seems unlikely)? The paper makes an unsubstantiated that the notion applies to any combinatorial problem without specifying what is meant by a combinatorial problem either. Note that many algorithms with predictions don't actually predict solutions but properties of the input, e.g. the degree of a vertex in a graph.


Solely considering the results and proofs for facility location and set cover, the paper is unfortunately written in a hand-wavy matter with informal definition and proofs. As written, it is impossible for me to verify that the results are  correct (I will provide a non-exhaustive list of examples below in my questions for the authors).

As such I think that the notion of discrete-smoothness could be relevant and that it touches upon important areas of research within learning-augmented algorithms. However, without a proper rigorous definition of discrete-smoothness and with a serious improvement in the write-up of the theorems, definitions, and proofs, I don't think the paper is in a state to be accepted.

**Questions:**

l32: For a general combinatorial problem, it is very unclear what $OPT\cap S$ means mathematically. Consider e.g, a matching problem where $S$ consists of  prediction of the degree of a node in a graph. Or the number of triangles that an edge is contained in for triangle counting. Notions like $OPT|_{S}$ seem to be overloaded with meaning (both $OPT\cap S$ but also the cost of $OPT$ on $S$ (whatever that means for general problems)).

l87: What is $m$?

l123-124: What does 'against the unconstrained optimum OPT' mean?

l126: What does it mean that $OPT|_S $ 'serves' a request? Is $OPT|_S$ part of the optimum solution? Is it a number? Same for l130

l147: The definition of $ALG(Q'|Q)$ is unclear. It is unclear what it means that the algorithm addresses a subset of the input.

l195: What does it mean that it has a 'monotone online amortization'? Hasn't been introduced and doesn't sound standard. What does it mean that it is $O(\log |S|)$-subset competitiveusing OA w.r.t. S?

l211: Theorem 2.3 is about the \emph{expected} distance over the distribution. Do you pick a random tree to embed into?

l232: What is $c(e)$?

Observation 2.4: Should it have a proof? Is it easy? It seems correct but with the writing so unclear it's hard for me to tell.

l252-254: What does it mean that it 'climbs up the branch', 'advances the proxy list', 'invests a growing amount'? Please, be more precise.

Algorithm 3: The while loop doesn't increase the $x_s$. Should it be an if and just increase $y_q$ to $\pi$?

Lemma 3.3: Has $ALG(q,\pi)$ been defined?

Lemma 3.4: How is 'the non-prize-collecting input formed from $Q'$' defined?

l301: Line 11 doesn't seem to have a return.

l329-330: What is the standard online algorithm without predictions? What is the online algorithm restricted to the predicted set?

**Limitations:**

None as far as I can tell

---

> ### Author Rebuttal · Authors · 2023-08-09
>
> We thank the reviewer for their comments. Here is our response:
>
> l32: We refer the reviewer to the example in the general rebuttal, which exemplifies what we meant by combinatorial problems, and how this notion captures the (min-cost, bipartite) matching problem. Note that in our paper we consider predictions which imply a set of variables (rather than the degree of a node). That is, a prediction would imply a collection of “recommended” matches.
>
>
> l87: $m$ is the number of points in the metric space. We see that it is mistakenly only defined in Line 192; we will add the definition to the introduction as well.
>
> l123-l124: The (unconstrained) optimal solution can use all of the items in the input. This is in contrast to the optimal solution that is restricted to using only predicted items, which is referred to in the previous sentence (Line 122).
>
> l126: The term $OPT|_S$ is overloaded to refer to both the set of items $OPT\cap S$ and the cost of this set of items. While this notation overload is very common (e.g., for $OPT$ itself), we will modify the paper to make it explicit.
>
>
> l147: The input $Q$ is a sequence of requests that arrives one after the other. Upon each request, the algorithm must buy items such that the request is satisfied; this is what we call “addressing” the request. $ALG(Q’|Q)$ is thus the total cost incurred when addressing the requests of the subset $Q’$.
>
> l195: Online amortization refers to using an amortized cost for the steps taken by the online algorithm, rather than the actual cost. For some problems, such as facility location, subset competitiveness can only be shown w.r.t. this amortized cost.
> Specifically, when the algorithm opens a new, expensive facility upon some request $q$, this is not subset-competitive relative to the singleton request set containing $q$.
> Thus, a way of “spreading out” the cost of opening facilities over multiple requests is required – and this is achieved by online amortization.
> Online amortization has also been used by [9] for a similar purpose.
>
> l211: Yes, the embedding is randomized, drawing a random tree from the distribution $\mathcal{D}$ described in Theorem 2.3.
>
> l232: As defined in Line 212, $c(e)$ is the cost of edge $e$ in the given tree.
>
>
>
> Observation 2.4: The procedure in Lines 226-229 explicitly constructs a Pareto frontier from the leaves under $v$, in terms of the logarithmic class of facility opening cost and the logarithmic class of distance from $v$. In Observation 2.4, the first item simply states that every leaf $u$ is dominated by some element of the frontier; that is, there is a point in the frontier where the class opening cost is at most $\ell(f_u)$, and the class of connection cost is at most $\ell(\delta_{u, v})$. Properties 2,3 of Observation 2.4 simply refer to the ordering chosen for the frontier (increasing facility cost, decreasing distance from $v$).
>
> l252-l254: This intuitive, verbal description of the algorithm is meant to accompany the formal description, which appears in Algorithm 2 as pseudocode.
>
> Algorithm 3: Note that the “foreach” loop of Line 10, which does increase the $x_s$ variables, is included in the “while” loop.
>
> Lemma 3.3: $ALG(q, \pi)$ simply refers to the cost of the algorithm when addressing the request $(q,\pi)$; we will add this definition to the paper.
>
> Lemma 3.4: The non-prize-collecting input is formed from the prize-collecting input by using the same requests but without the option of paying penalties. We’ll clarify this in the paper.
>
> l301: The line references here seem to have been shifted by one, this reference actually refers to Line 12. This is also the case in Line 303, in which the reference is to Line 9 in the algorithm rather than Line 8.
>
> l329-l330:  The standard online algorithm without predictions refers to the algorithm of Theorem 3.1. The online algorithm restricted to the predicted set refers to the algorithm of Corollary 3.2. Equivalently, these are instances of the standard multiplicative-update algorithm for set cover, where all sets are allowed (Theorem 3.1) or when the algorithm discards unpredicted sets (Corollary 3.2).

---

### Official Review · Reviewer_MB7r · 2023-07-05

**Soundness:** 4 excellent
**Presentation:** 3 good
**Contribution:** 4 excellent
**Rating:** 8
**Confidence:** 2

**Summary:**

This paper presents the concept of "discrete smoothness" in the context of online algorithms with predictions. This notion smoothly bridges the gap between consistency and robustness, terms originally introduced by Purohit et al. For a class of problems called "online covering problem", the authors first show that a "Lagrangian subset-competitive algorithm" for the prize-collecting version of the problem can be transformed into an algorithm with discrete smoothness for the original problem. The paper then proposes Lagrangian subset-competitive algorithms for the prize-collecting versions of online facility location and online set cover problems, which are special cases of online covering problems. The effectiveness of the proposed algorithm is substantiated through numerical experiments.

**Strengths:**

Though I am not an online algorithms expert, I feel this paper includes many nontrivial arguments. The coined notion (discrete smoothness) is natural (except for the name; see Questions below). It is quite intriguing that a Lagrangian subset-competitive algorithm for the prize-collecting version can be transformed into an algorithm with discrete smoothness. Its proof is not straightforward. The algorithms for online facility location and set cover are also solid.

**Weaknesses:**

I was unable to identify any significant weaknesses.

**Questions:**

- Can you devise a Lagrantian subset-competitive algorithm for a general online covering problem?
- The appropriateness of the term *discrete smoothness* is questionable. Several papers discuss "smoothness" for discrete problems and algorithms (without predictions; see, e.g., https://arxiv.org/abs/2211.04674). The concept you've proposed appears to be more focused on predictions rather than the discreteness of the problem. However, "discrete smoothness" doesn't inherently imply any association with predictions.

**Limitations:**

Yes

---

> ### Author Rebuttal · Authors · 2023-08-09
>
> We thank the reviewer for their comments. Here is our response:
>
> * We are unaware of a general construction that can encapsulate any algorithm for a covering problem and add the subset-competitiveness or Lagrangian properties.
> * The term “smoothness” has often been used in online algorithms with predictions to refer to a graceful degradation of performance as a function of prediction error (e.g., in [11]). We added the word “discrete” to differentiate between the smoothness guarantee we provide and existing, continuous notions of smoothness (e.g., where the prediction error involves continuous amounts, such as distances in a metric space). We appreciate the input regarding the naming choice, and will take it into account.

---

> > ### Comment · Reviewer_MB7r · 2023-08-18
> >
> > Thanks for the comments!

---

### Author Rebuttal · Authors · 2023-08-09

We would like to thank the reviewers for their thorough reviews and thoughtful comments. We would like to clarify some aspects of the discrete-smoothness notion and its broad applicability.
The structure of many combinatorial problems is the following: one must assign binary values in {0,1} to a set of variables, such that the assignment satisfies some constraints, and incurs a cost which is a function $f$ of the variables to which 1 is assigned. Suppose one is given a prediction which induces any subset $S$ of these variables; then, the notion of discrete-smoothness in Equation (1) yields a bound which is a function of the affinity between OPT (the optimal subset of variables to which 1 is assigned) and the predicted subset $S$. Note the following:
Discrete-smoothness does not dictate how the subset $S$ induced by the prediction is mediated to the algorithm. It could be that $S$ is simply given to the algorithm in advance; but, the prediction could also be a rule mapping a given request to the variables that should serve it, in which case $S$ is the union of suggested variables over the run of the algorithm. (For example, in set cover, the prediction could map an arriving element to the set that should be bought to address it.)
Discrete-smoothness does not demand that the subset induced by $S$ be feasible for the constraints induced by the input. Indeed, a possible use case in a real-world scenario would be to use a predictor that generates a solution for 90% of the arriving online input.

Here is a concrete example, outside the scope of our paper, to which this notion could apply. In min-cost bipartite matching, the variables are of the form $x_{u,v}$, denoting the matching of node u with node v. A prediction would imply a set $S$ of possible pairs (not necessarily a feasible matching), and can be delivered, e.g., through a recommendation upon the arrival of a node (“match this node with one of $v_1, v_2, v_3$”).
Another concrete example is Steiner tree, in which each variable represents buying an edge in the graph.

---

### Decision · Program_Chairs · 2023-09-21

**Decision:**

Accept (poster)

**Comment:**

The reviewer are divided regarding this paper. Reviewer h71T provides strong support for the paper's contributions and techniques. In contrast, Reviewers 9LeQ and mT48 are concerned about the presentation and the significance of the discrete smoothness definition introduced in the paper to bridge robustness and consistency in online optimization with learned predictions. Reviewer mT48 is also concerned about the novelty of the algorithms for the two main concrete problems addressed (set cover and facility location). The discrete smoothness definition is a new conceptual contribution that may lead to further progress. Additionally, the improvements obtained for the two problems considered (set cover and facility location) and the techniques involved are valuable contributions as well. For these reasons, I recommend accepting the paper.